# Zero-Shot Robotic Manipulation With Pretrained Image-Editing Diffusion Models

**Kevin Black**[*1]  **Mitsuhiko Nakamoto**[*1]  **Pranav Atreya**[1]  **Homer Walke**[1]
**Chelsea Finn**[2,3]  **Aviral Kumar**[1,3]  **Sergey Levine**[1,3]

[1]University of California, Berkeley  [2]Stanford University  [3]Google DeepMind
[*]Equal contribution

## Abstract

If generalist robots are to operate in truly unstructured environments, they need to be able to recognize and reason about novel objects and scenarios. Such objects and scenarios might not be present in the robot's own training data. We propose SuSIE, a method that leverages an image-editing diffusion model to act as a high-level planner by proposing intermediate subgoals that a low-level controller can accomplish. Specifically, we finetune InstructPix2Pix on video data, consisting of both human videos and robot rollouts, such that it outputs hypothetical future "subgoal" observations given the robot's current observation and a language command. We also use the robot data to train a low-level goal-conditioned policy to act as the aforementioned low-level controller. We find that the high-level subgoal predictions can utilize Internet-scale pretraining and visual understanding to guide the low-level goal-conditioned policy, achieving significantly better generalization and precision than conventional language-conditioned policies. We achieve state-of-the-art results on the CALVIN benchmark, and also demonstrate robust generalization on real-world manipulation tasks, beating strong baselines that have access to privileged information or that utilize orders of magnitude more compute and training data. The project website can be found at http://rail-berkeley.github.io/susie.

## 1 Introduction

A useful generalist robot must be able to — much like a person — recognize and reason about novel objects and scenarios it has never encountered before. For example, if a user instructs the robot to "hand me that jumbo orange crayon," it ought to be able to do so even if it has never interacted with a jumbo orange crayon before. In other words, the robot needs to possess not only the physical capability to manipulate an object of that shape and size but also the semantic understanding to reason about an object outside of its training distribution. As much as robotic manipulation datasets have grown in recent years, it is unlikely that they will ever include every conceivable instance of objects and settings, any more so than the life experiences of a person ever include physical interactions with every type of object. While these datasets contain more than enough examples of manipulating elongated cylindrical objects, they lack the broad semantic knowledge necessary to ground the *particular* objects that robots will encounter during everyday operation.

How can we imbue this semantic knowledge into language-guided robotic control? One approach would be to utilize models pretrained on vision and language to initialize different components of the robotic learning pipeline. Recent efforts, for example, initialize robotic policies with pretrained vision-language encoders [7] or apply pretrained models to semantic scene augmentation [11, 64]. While these methods bring semantic knowledge into robot learning, it remains unclear if these approaches realize the full potential of Internet pretraining in improving robotic policy execution at every level, or whether they simply improve the high-level visual generalization of the policy.

We propose an approach for leveraging pretrained image-editing models to enable generalizable robotic manipulation. We finetune the image-editing model on video data such that, given the current frame and a language description of the current task, the model generates a hypothetical *future*

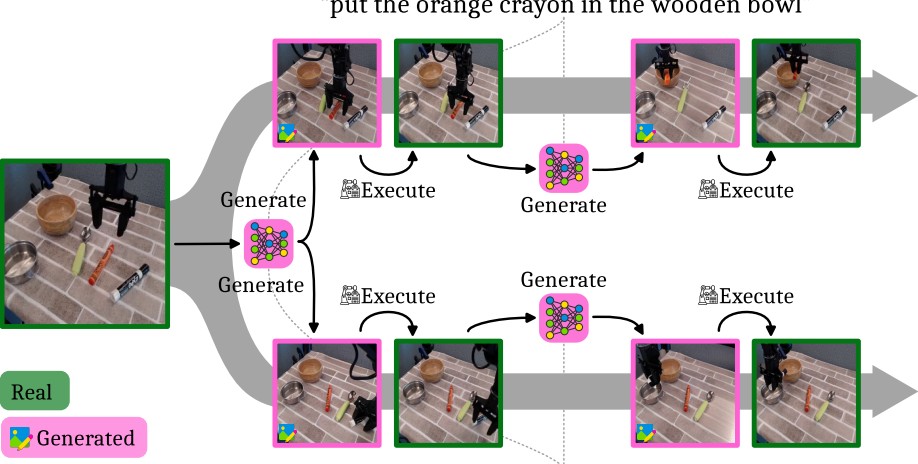

**Figure 1: (SuSIE)** Our method leverages a pretrained image-editing model to generate future subgoals based on language commands. A low-level goal-reaching policy then executes the actions needed to reach each subgoal. Alternating this loop enables us to solve the task.

frame. This does not require the model to precisely understand the intricacies of the robot's low-level dynamics and therefore should facilitate transfer from other data sources (e.g., human videos) where the low-level physical interactions and precise embodiment do not match. At test time, we employ a low-level goal-reaching policy trained on robot data to reach this hypothetical future frame; this policy, in turn, only needs to infer visuo-motor relationships to determine the correct actuation and does not need to understand the overlying semantics. Furthermore, such subgoals can simplify the task by inferring likely poses for the arm at intermediate substeps, such as the pose corresponding to grasping an object (see Figure 1). In fact, we observe in our experiments that even when existing approaches possess sufficient semantic understanding to solve a task, they often fail due to imprecise localization of obstacles and objects; following the generated subgoals enables our method to perform well in such scenarios. Much like a person first constructs a high-level plan to complete a task before deferring to muscle memory for the low-level control, our method can also be viewed as first running a high-level planner with integrated semantic reasoning *and* visual understanding before deferring to a low-level controller to execute the plan.

The main contribution of our work is **SU**bgoal **S**ynthesis via **I**mage **E**diting (**SuSIE**), a simple and scalable method for incorporating semantic information from pretrained models to improve robotic control. A pretrained image-editing model is used with minimal modification, requiring only finetuning on video data. The low-level goal-conditioned policy is trained with standard supervised learning, and faces the comparatively easy problem of reaching nearby image subgoals; this typically only requires attending to a single object or the arm position, ignoring most parts of the scene. Together, this approach achieves state-of-the-art results on the CALVIN benchmark and solves real robot control tasks involving novel objects, novel distractors, and even novel scenes. It outperforms all prior approaches on these real-world tasks, including oracle baselines with privileged information and RT-2-X [14] — a 55 billion parameter model trained on Internet-scale vision-language data as well as an order of magnitude more robot data than SuSIE.

## 2   RELATED WORK

**Incorporating semantic information from pretrained vision-language models.** Prior works that incorporate semantic information from pretrained vision-language models (VLMs) into robot learning can be classified into two categories. The first category aims to improve visual scene understanding in robot policies with information from VLMs. For instance, GenAug [11], ROSIE [64], DALL-E-Bot [34], and CACTI [43] use text-to-image generative models to produce semantic augmentations of a given scene with novel objects and arrangements and train the robot policy on the augmented data to enable it to perform well in a similar scene. MOO [57] uses a pretrained object detector to extract bounding boxes to guide the policy towards the target object. Other works di-

rectly train language and image-based policies [6, 7, 55], some using VLM initializations [15, 52], to produce action sequences.

While these approaches do utilize pretrained models, we find in our experiments that pretraining using VLMs [7] does not necessarily enhance low-level policy execution in the sense that learned policies often localize the object or move the gripper imprecisely (see Figure 3). On the other hand, our approach is able to incorporate benefits of pretraining into low-level control by synthesizing subgoals that carefully steer the motion of the low goal reaching policy, improving its precision. While our approach can be directly applied in *unstructured* open-world settings, applying methods from [11, 64, 43] requires additional prior information, such as object bounding boxes or 3D meshes. This significantly restricts their applicability to scenarios where this additional information is not available: for example, GenAug [11] is not applicable in our experiments since 3D object meshes for new target objects were not available. Distinct from our approach for utilizing generative models, other works design representation learning objectives for vision-language pretraining for control [49, 42, 35, 4], but these methods still need to utilize limited amounts of data from the target task to learn a policy. In contrast, our approach solves the target task in zero-shot, without any task-specific data.

The second category of approaches incorporates semantic information from pretrained models for informing low-level controllers. Typically, these prior methods use pretrained models to imagine visual [16, 2] or textual plans [8, 28, 29, 39], which then inform a low-level robot control policy. Of these, methods that train a low-level policies conditioned on text suffer from a grounding problem, which our approach circumvents entirely since the low-level policy only sees images.

Perhaps the most related methods to our approach in this second category are UniPi [16] and HiP [2], which train video models to generate a sequence of frames achieving the target task, and then extract robot actions by training an inverse dynamics model. Our approach does *not* attempt to generate full videos (i.e., *all* frames in a rollout), but only the next waypoint that a low-level policy must achieve to solve the commanded task. While this difference might appear small, it has major implications: modeling an entire video puts a very high burden on the generative model, requiring the frames to obey strict physical consistency. Unfortunately, we find that open-source video models often produce temporally inconsistent frames, which confuse the low-level controller (see Appendix C). Our method provides more freedom to the low-level controller to handle the physical aspects of the task over a longer time interval while guiding it at a level that is suitable to the diffusion model's ability to preserve physical plausibility. In our experiments, we find that our method significantly improves over an open-source reimplementation of UniPi [16].

**Classical model-based RL and planning with no pretraining.** The idea behind our approach is also related to several methods in the deep RL literature that do not use pretrained models and generally do not study language-guided control. For instance, [21, 37, 63, 60, 53, 22] train action-conditioned dynamics models and run RL in the model. While our approach also models multi-step dynamics, our model is not conditioned on an action input. Removing the dependency on an action input enables us to de-couple the finetuning of the (large) image-editing model from the policy entirely, improving simplicity and time efficiency. APV [54] trains an action agnostic dynamics model from videos but finetunes it in a loop with the policy, and hence, does not enjoy the above benefits. Finally, these model-based RL methods do not exhibit zero-shot generalization abilities to new tasks, which is an important capability that our method enjoys. Our approach is also related to several video prediction methods [17, 38, 3, 58] but utilizes a better neural network architecture (i.e., diffusion models instead of LSTMs and CNNs). Most related is to our method is hierarchical visual foresight (HVF) [48]: while HVF uses MPC to find an action, our approach uses a goal-reaching policy thereby eliminating the cost of running MPC with large dynamics models.

Our approach is also related to prior works that use generative models for planning in a single-task setting, with no pretraining. Trajectory transformer (TT) [31], decision transformer (DT) [10], and their extensions condition the policy on the target return or goal. While diffusion-based variants of these methods [32, 1] use diffusion models like our approach, they still require training data from the target task to learn a policy, unlike our zero-shot planning approach.

## 3 PRELIMINARIES AND PROBLEM STATEMENT

We consider the problem setting of language-conditioned robotic control. Specifically, we want a robot to accomplish a task described by a novel language command. We divide sources of data into 3 categories: language-labeled video clips $\mathcal{D}_l$ that do not include robot actions; language-labeled

robot data $\mathcal{D}_{l,a}$ that include both language labels and robot actions; and unlabeled robot data $\mathcal{D}_a$ that include only actions (e.g., play data).

Formally, we define $\mathcal{D}_{l,a} = \{(\tau^1, l^1), (\tau^2, l^2), \cdots, (\tau^N, l^N)\}$, where each trajectory $\tau^n$ consists of a sequence of images (or states) $\mathbf{s}_i^n \in \mathcal{S}$ and actions $\mathbf{a}_i^n \in \mathcal{A}$ that were executed while collecting this data, i.e. $\tau^n = (\mathbf{s}_0^n, \mathbf{a}_0^n, \mathbf{s}_1^n, \mathbf{a}_2^n, \cdots)$, following the standard assumptions of a Markov decision process. $l^n$ is a natural language command describing the task accomplished in the trajectory. $\mathcal{D}_l$ and $\mathcal{D}_a$ are organized similarly, but are missing either actions $\mathbf{a}_i^n$ or language annotations $l^n$, respectively. At test time, given a new scene $\mathbf{s}_0^{\text{test}}$ and a new natural language description $l^{\text{test}}$ of a task, we evaluate a method in terms of its success rate at accomplishing this task starting from $\mathbf{s}_0^{\text{test}}$.

## 4 SuSIE: Subgoal Synthesis via Image Editing

Our goal is to utilize semantic information from the Internet to improve language-guided robot control in the presence of novel environments, scenes, and objects. How can we do this when models trained on general-purpose Internet data do not provide guidance in selecting low-level actions? Our key insight is that we can effectively leverage the capabilities of the pretrained model if we decouple the robot control problem into two phases: **(I)** generating subgoals that would need to be attained to succeed at the task, and **(II)** learning low-level control policies for reaching these generated subgoals. Our method incorporates semantic information from both text-image pretraining on Internet data as well as non-robot video data in Phase (I) by finetuning a text-guided image-editing model on $\mathcal{D}_l \cup \mathcal{D}_{l,a}$. Phase (II) is accomplished via a goal-conditioned policy trained on the robot data $\mathcal{D}_{l,a} \cup \mathcal{D}_a$. We describe each of these phases below and then summarize the resulting algorithm.

### 4.1 Phase (I): Synthesizing Subgoals From Image-Editing Models

The primary component of our method is a generative model that, given a target task specified by natural language, can guide a low-level controller to a state that advances the task. One way to accomplish this is to train a generative model to produce an immediate next subgoal image. We can then incorporate semantic information from the Internet into our algorithm by initializing this generative model with a suitable pretrained initialization, followed by finetuning it on multi-task, diverse video data consisting of robot rollouts as well as other videos from the Internet.

*What is a good pretrained initialization for this model?* Our intuition is that, since accomplishing a task is equivalent to "editing" the pixels of an image of the robot workspace under constraints prescribed by the language command, a good pretrained initialization may be provided by a text-guided image-editing model. We instantiate our approach with Instruct Pix2Pix [9], though other image-editing models could also be used. Formally, this model is given by $p_\theta(\mathbf{s}_{\text{edited}} \mid \mathbf{s}_{\text{orig}}, l)$. Then, using the dataset $\mathcal{D}_l \cup \mathcal{D}_{l,a}$ of language-labeled video clips and robot trajectories, we finetune $p_\theta$ to produce valid subgoals $\mathbf{s}_{\text{edited}}$ given an initial image $\mathbf{s}_{\text{orig}}$ and a language label $l$. Formally, the training objective is given by

$$\max_\theta \ \mathbb{E}_{(\tau^n, l^n) \sim \mathcal{D}_l \cup \mathcal{D}_{l,a}; \ \mathbf{s}_i^n \sim \tau^n; \ j \sim q(j|i)} \left[ \log p_\theta\left(\mathbf{s}_j^n \mid \mathbf{s}_i^n, l^n\right) \right], \tag{1}$$

where $q(j \mid i)$ is a distribution of our choosing that controls what subgoals the model is trained to produce. We want the diffusion model to generate subgoals that are a consistent distance in the future from the current state — close enough to be reachable by the low-level policy, but far enough to achieve significant progress on the task. As such, we choose dataset-dependent hyperparameters $k_{\min}$ and $k_{\max}$, and set $q$ to select subgoals uniformly between $k_{\min}$ and $k_{\max}$ steps in the future, i.e.,

$$q(j \mid i) = U\left(j; \ [i + k_{\min}, i + k_{\max})\right).$$

### 4.2 Phase (II): Reaching Generated Subgoals with Goal-Conditioned Policies

In order to utilize the finetuned image-editing model to control the robot, we additionally need to train a low-level controller to select suitable robot actions. In this section, we present the design of our low-level controller, followed by a full description of our test-time control procedure. Since the image-editing model in SuSIE produces images of future subgoals conditioned on natural language task descriptions, our low-level controller can simply be a language-agnostic goal-reaching policy.

**Training a goal-reaching policy.** Our goal-reaching policy is parameterized as $\pi_\phi(\mathbf{a} \mid \mathbf{s}_i, \mathbf{s}_j)$, where $\mathbf{s}_j$ is a future frame that the policy intends to reach by acting at $\mathbf{s}_i$. At test time, we only

need the low-level policy to be proficient at reaching close-by states that lie within $k_{\max}$ steps of the current state, since the image-editing model from Phase (I) is trained to produce subgoals within $k_{\max}$ steps of any state. To train this policy, we run goal-conditioned behavioral cloning (GCBC) on the robot data $\mathcal{D}_{l,a} \cup \mathcal{D}_a$. Formally, the training objective is given by

$$\max_{\phi} \; \mathbb{E}_{\tau^n \sim \mathcal{D}_{l,a} \cup \mathcal{D}_a; \; (\mathbf{s}_i^n, \mathbf{a}_i^n) \sim \tau^n; \; j \sim U([i, i+k_{\max}+k_\delta])} \left[ \log \pi_\phi(\mathbf{a}_i^n \mid \mathbf{s}_i^n, \mathbf{s}_j^n) \right], \qquad (2)$$

where $k_\delta$ is another hyperparameter that provides a small amount of overhead, since the image-editing model is not perfect and may not always produce subgoals that are reachable in less than $k_{\max}$ steps, especially for unseen tasks.

**Test-time control with $\pi_\phi$ and $p_\theta$.** Once both the goal-reaching policy $\pi_\phi$ and the subgoal generation model $p_\theta$ are trained, we utilize them together to solve new manipulation tasks based on user-specified natural language commands. Given a new scene $\mathbf{s}_0^{\text{test}}$ and a language command $l^{\text{test}}$, SuSIE attempts to solve the task by iteratively generating subgoals and commanding the low-level policy to reach these subgoals. At the start, we sample the first subgoal $\widehat{\mathbf{s}}_+ \sim p_\theta(\mathbf{s}_+ \mid \mathbf{s}_0^{\text{test}}, l^{\text{test}})$. Once the subgoal is generated, we then roll out the goal-reaching policy $\pi_\phi$ conditioned on $\widehat{\mathbf{s}}_+$ for $k_{\text{test}}$ timesteps where $k_{\text{test}}$ is a test-time hyperparameter. After $k_{\text{test}}$ timesteps, we refresh the subgoal by sampling from the subgoal generation model again and repeat the process. In practice, for computational efficiency, we set $k_{\text{test}}$ to be similar to the corresponding $k_{\max}$ used with the robot data and found this to be sufficient for obtaining good performance. However, given an unlimited computational budget, conventional wisdom would suggest that regenerating subgoals more often would lead to more robust control. Pseudocode for test-time control is provided in Algorithm 1.

---

**Algorithm 1** SuSIE: Zero-Shot, Test-Time Execution

---

**Require:** Subgoal model $p_\theta(\mathbf{s}_+ \mid \mathbf{s}_t, l)$, policy $\pi_\phi(\mathbf{a} \mid \mathbf{s}_t, \mathbf{s}_+)$, time limit $T$, subgoal sampling interval $k_{\text{test}}$, initial state $\mathbf{s}_0^{\text{test}}$, language command $l^{\text{test}}$
1: $t \leftarrow 0$
2: **while** $t \leq T$ **do**
3:     Sample $\widehat{\mathbf{s}}_+ \sim p_\theta(\mathbf{s}_+ \mid \mathbf{s}_t^{\text{test}}, l^{\text{test}})$         ▷ Generate a new subgoal every $k_{\text{test}}$ steps
4:     **for** $j = 1$ to $k_{\text{test}}$ **do**
5:         Sample $\mathbf{a}_t \sim \pi_\phi(\mathbf{a} \mid \mathbf{s}_t^{\text{test}}, \widehat{\mathbf{s}}_+)$    ▷ Predict the action from current state and subgoal
6:         Execute $\mathbf{a}_t$
7:         $\mathbf{s}_{t+1}^{\text{test}} \leftarrow$ robot observation
8:         $t \leftarrow t + 1$
9:     **end for**
10: **end while**

---

### 4.3 IMPLEMENTATION DETAILS

In Phase (I), we utilize the pretrained initialization from the InstructPix2Pix model [9]. We implement Equation 1 using the standard variational lower bound objective for training diffusion models [25]. Our diffusion model and policy operate on images of size $256 \times 256$. To ensure that this model pays attention to the input image and the language command, we apply classifier-free guidance [24] separately to both the language and the image, following InstructPix2Pix. To obtain a robust goal-reaching policy in Phase (II), we use a diffusion policy [12, 23] that predicts chunks of 4 actions and performs temporal averaging over these predictions at test time [65]. More details about the training hyperparameters and architecture are provided in Appendix A.1.

## 5 EXPERIMENTAL EVALUATION

The goal of our experiments is to evaluate the efficacy of SuSIE at improving generalization and the low-level policy execution in open-world robotic manipulation tasks. To this end, our experiments aim to study the following questions:

1. Can SuSIE solve a task in a novel environment, with novel objects, and with a novel language command, in zero-shot?

2. Does SuSIE exhibit an elevated level of precision and dexterity compared to other approaches that do not use subgoals?

3. How crucial is pretraining on Internet data, as well as cotraining on non-robot video data, for attaining zero-shot generalization?

To answer these questions, our experiments compare SuSIE to several prior methods including state-of-the-art approaches for training language-conditioned policies that leverage pretrained vision-language models in a variety of ways.

## 5.1 EXPERIMENTAL SCENARIOS AND COMPARISONS

**Real-world experimental setup and datasets.** We conduct our real-robot experiments on a WidowX250 robot platform. Our dataset is BridgeData V2 [59], a large and diverse dataset of robotic manipulation behaviors designed for evaluating open-vocabulary instructions. The dataset contains over 60k trajectories, 45k of which are language-labeled, which we use as our language-labeled robot dataset $\mathcal{D}_{l,a}$. We use the remaining 15k trajectories as our action-only dataset $\mathcal{D}_a$.

Our video-only dataset $\mathcal{D}_l$ is the Something-Something dataset [19], a dataset consisting of short video clips of humans manipulating various objects. We chose Something-Something because it primarily contains examples of object manipulation with a still camera frame and hence exhibits a smaller domain gap to the robot data collected with an over-the-shoulder camera compared to other video datasets that contain substantial egocentric motion. We filter the Something-Something dataset using its labels to remove trajectories that contain egocentric motion or otherwise lack significant manipulation behavior, producing a final dataset size of approximately 75k video clips.

Our evaluations present three different scenes (Figure 2) designed specifically to test the ability of various methods at different levels of open-world generalization: **Scene A:** this scene includes an environment and objects that are well-represented in BridgeData V2; **Scene B:** this scene is situated in an environment with a seen tabletop but a novel background and distractors, where the robot must move a seen object (bell pepper) into a choice of seen container (orange pot) or unseen container (ceramic bowl); and **Scene C:** this scene includes a table texture unlike anything in BridgeData V2 and requires manipulating both seen and unseen objects. Semantically, Scene C is the most challenging since the robot needs to carefully ground the language command to identify the correct object while resisting its affinity for an object that is well-represented in the robot data (the spoon). Scene B also requires semantic grounding to distinguish between the seen and unseen receptacles, while adding the additional challenge of manipulating the plastic bell pepper — this object requires particular precision to grasp due to being light, smooth, and almost as wide as the gripper.

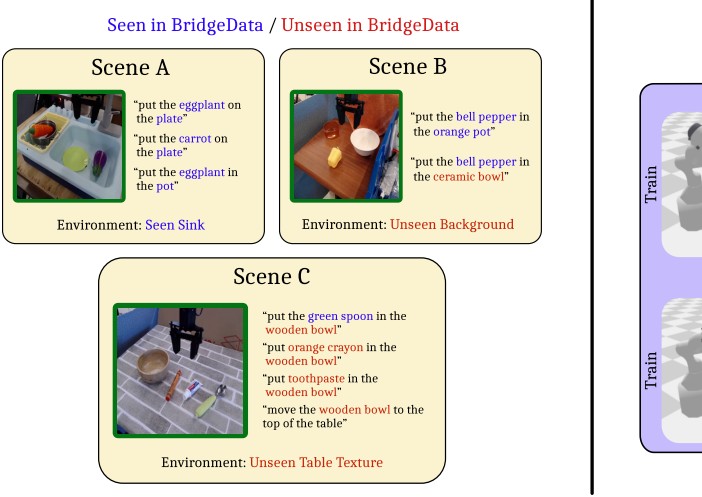
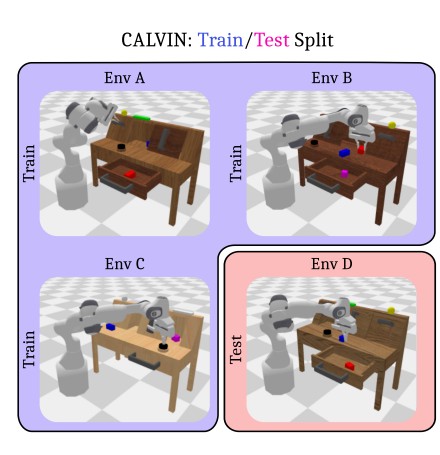

**Figure 2: (Experimental setup) (L)** We evaluate our method on 9 tasks across 3 real-world scenes. In terms of semantic generalization, the scenes become progressively more difficult, due to both an increasing visual departure from the robot training data and an increasingly confounding mixture of both seen and unseen objects. **(R)** In simulation, we evaluate our method in the zero-shot setting of the CALVIN benchmark, which involves training on 3 environments (A, B, and C) and testing on a 4th environment (D). The environments differ in table texture, positioning of furniture elements, and possible configurations of various colored blocks. Each environment comes with 34 language-specified tasks.

**Simulation tasks.** We run our simulation experiments in CALVIN [45], a benchmark for long horizon, language-conditioned manipulation. CALVIN consists of four simulated environments, labeled A through D, and each environment comes with a dataset of human-collected play trajectories. Approximately 35% of the play data is annotated with language, which we use as our language-labeled robot dataset $\mathcal{D}_{l,a}$. We use the remaining 65% of the play data as our action-only robot dataset $\mathcal{D}_a$, and do not use any video-only dataset $\mathcal{D}_l$ in our simulation experiments.

Each environment consists of a Franka Emika Panda robot arm positioned next to a desk with various manipulatable objects, including a drawer, sliding cabinet, light switch, and various colored blocks. Environments are differentiated by their table textures, positions of the furniture objects, and configurations of colored blocks. With this benchmark, we study the most challenging zero-shot multi-environment scenario: training on A, B, and C, and testing on D. We follow the evaluation protocol from Mees et al. [45]. During evaluation, a policy is given 360 timesteps to complete a chain of five language instructions. We provide more details about our experimental setup in Appendix B.

**Comparisons.** Our experiments cover methods that utilize pretrained models of vision and language in language-guided robot control in a variety of ways. Several prior methods tackle language-based robotic control as discussed in Section 2. In our experiments, we choose to compare to a representative subset of these prior methods to maximally cover the possible set of comparisons. In our real-world experiments, we compare to the following methods:

- **RT-2-X** [14], a 55 billion parameter vision-language model finetuned on the full Open X-Embodiment dataset to produce robot actions. The robot training data includes all of Bridge-Data V2, as well as a vast quantity of additional robot manipulation data totaling over 1.1 million trajectories.

- **MOO** [57], which utilizes pretrained object detectors to obtain 2D localization information for the manipulation targets and trains a language-conditioned behavioral cloning (LCBC) policy with this extra information. We re-implement MOO by using a large language model to extract the manipulation targets from the unstructured language labels in BridgeData V2, then OWL-ViT [46] to extract 2D bounding boxes for these objects. Unlike MOO, at test time we manually give the policy the ground truth 2D bounding boxes of the manipulation targets.

- **UniPi** [16], which finetunes a language-conditioned video prediction model on robot data. Since the original UniPi model utilized proprietary pretrained initializations that are not available publicly, we replicated this method in two different ways: (1) using the UniPi model from HiP [2], which is a PVDM [62] latent video diffusion model, and (2) by implementing our own image-space video diffusion model with a factorized spatial-temporal 3D UNet similar to Ho et al. [27]. We found that (1) was not able to produce reasonable-looking videos in the real robot scene (Appendix C), so we only evaluated (2) in our real-world experiments.

- **LCBC**, which trains a low-level policy conditioned on the language labels in BridgeData V2. This method is broadly representative of prior language-conditioned behavioral cloning methods such as RT-1 [6] and BC-Z [30]. We use the same architecture and hyperparameters as the low-level policy in SuSIE, as we found that they outperformed RT-1 on the BridgeData tasks.

We provide more details about the baselines and their implementations in Appendix A.2.

In our simulation experiments, we compare to additional methods previously studied on CALVIN. This includes methods that explicitly tackle long-horizon language-based control on CALVIN such as multi-context imitation (MCIL) [41], hierarchical universal language-conditioned policy (HULC) [44], and improved variants of HULC [18]. We also compare to other state-of-the-art methods from Ge et al. [18] that employ an identical training and evaluation protocol as our experiments, namely MdetrLC [33] and AugLC [50].

## 5.2 CAN SuSIE PERFORM ZERO-SHOT ROBOTIC MANIPULATION?

**Simulation results.** We present performance in Table 1, in terms of success rates (out of 1.0) for completing chains of language instructions. SuSIE obtains superior zero-shot performance (train A, B, C → test D) compared to the previous state-of-the-art. We find that our reimplementation of UniPi achieves nontrivial success, but is still significantly outperformed by SuSIE.

**Real-world results.** We present performance of real-world evaluations in Table 2. SuSIE achieves the best performance across the board, beating RT-2-X, a 55 billion parameter model trained on

| | Success Rate for No. of Instructions in a Row | | | | | Avg. No. of |
| --- | --- | --- | --- | --- | --- | --- |
| | 1 | 2 | 3 | 4 | 5 | Instructions Completed |
| HULC [44] | 0.43 | 0.14 | 0.04 | 0.01 | 0.00 | 0.62 |
| MCIL [41] | 0.20 | 0.00 | 0.00 | 0.00 | 0.00 | 0.20 |
| MdetrLC [18] | 0.69 | 0.38 | 0.20 | 0.07 | 0.04 | 1.38 |
| AugLC [18] | 0.69 | 0.43 | 0.22 | 0.09 | 0.05 | 1.43 |
| LCBC | 0.67 | 0.31 | 0.17 | 0.10 | 0.06 | 1.31 |
| UniPi (HiP) [2] | 0.08 | 0.04 | 0.00 | 0.00 | 0.00 | 0.12 |
| UniPi (Ours) [16] | 0.56 | 0.16 | 0.08 | 0.08 | 0.04 | 0.92 |
| SuSIE (Ours) | **0.87** | **0.69** | **0.49** | **0.38** | **0.26** | **2.69** |

Table 1: (**CALVIN benchmark performance**) SuSIE is able to chain together more instructions with a higher success rate than all prior methods in the zero-shot (train A, B, C → test D) setting. We provide an example rollout in Appendix D.

| | Task | LCBC | MOO | UniPi | RT-2-X | SuSIE (Ours) |
| --- | --- | --- | --- | --- | --- | --- |
| | Eggplant on plate | 0.9 | 0.4 | 0.0 | 0.3 | **1.0** |
| | Carrot on plate | 0.4 | 0.3 | 0.0 | 0.6 | **0.9** |
| Scene A | Eggplant in pot | 0.6 | **0.7** | 0.0 | 0.4 | **0.7** |
| | Average | 0.63 | 0.47 | 0.0 | 0.43 | **0.87** |
| | Bell pepper in pot | 0.1 | 0.0 | 0.0 | 0.0 | **0.5** |
| Scene B | Bell pepper in bowl | 0.3 | 0.1 | 0.1 | 0.0 | **0.5** |
| | Average | 0.20 | 0.05 | 0.05 | 0.00 | **0.50** |
| | Toothpaste in bowl | 0.0 | 0.0 | 0.0 | 0.5 | **0.6** |
| | Crayon in bowl | 0.0 | 0.0 | 0.0 | 0.9 | **1.0** |
| Scene C | Spoon in bowl | 0.1 | 0.3 | 0.1 | 0.7 | **0.9** |
| | Bowl to top | 0.6 | 0.1 | 0.1 | 0.9 | **1.0** |
| | Average | 0.18 | 0.10 | 0.05 | 0.75 | **0.88** |

Table 2: (**Real-world performance**) SuSIE achieves the best success rate across the board, demonstrating both high precision as well as the ability to generalize to novel environments, objects, and language commands.

significantly more robot and Internet data. As expected, all methods perform well in Scene A, which is well-represented in the robot data. SuSIE performs uniquely well in Scene B, as it is the only method that can consistently grasp the bell pepper.

In Scene C, while SuSIE still achieves the best performance, RT-2-X comes in a close second. We hypothesize that this is because — in contrast to Scene B — all of the objects in Scene C are easy to grasp. Therefore, the low-level precision of the policy is less important, which is the primary weakness of RT-2-X, as we discuss further in the next section. Qualitatively, we observed that the failure cases in Scene C for both SuSIE and RT-2-X were almost always imprecise manipulations (failed grasps or early dropping) rather than semantic misunderstanding; of the 4 objects, the toothpaste is the most difficult to grasp, which is why its success rate is the lowest. That is, both methods solve the semantic understanding component of the tasks, but SuSIE's improved low-level precision allows it to perform slightly better.

The performance of LCBC is as expected: since it is trained only on BridgeData V2, it has no way of grounding novel objects, and often puts the bell pepper in the wrong receptacle (in Scene B) or attempts to grasp the wrong object (in Scene C). Surprisingly, it can recognize the wooden bowl in Scene C, even though that exact object does not appear in the training data; we hypothesize that the wooden bowl is distinctive enough to recognize from the word "bowl" alone combined with the many other bowls in BridgeData V2. MOO underperforms expectations, even with its privileged test-time information in the form of ground-truth bounding boxes. We hypothesize that, despite our best attempts, the target objects and bounding boxes extracted at train time are noisy due to the unstructured nature of BridgeData V2 and therefore only serve to confuse the policy (see Appendix A.2.1). UniPi's poor performance is due to the video model often producing temporally inconsistent frames (see Appendix C).

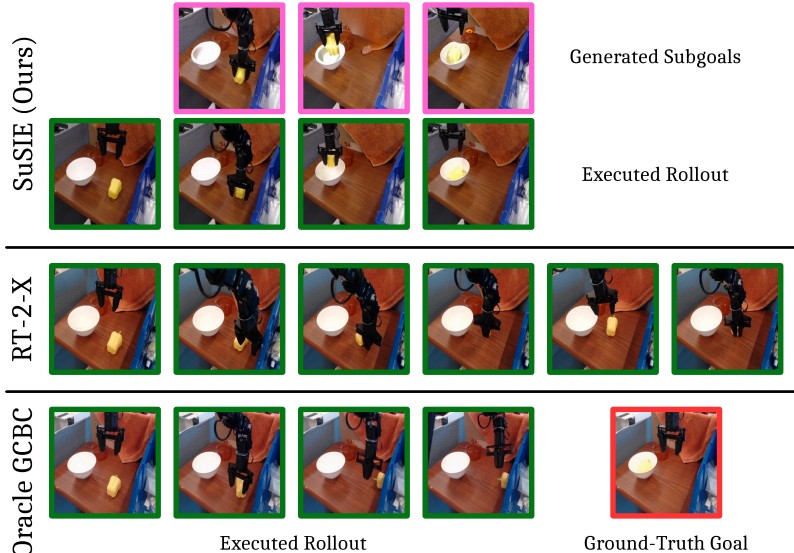

**Figure 3: (Visualizing rollouts)** Visualized rollouts from SuSIE, RT-2-X, and Oracle GCBC on the task "put the yellow bell pepper in the ceramic bowl" from Scene B. While RT-2-X and Oracle GCBC fail to grasp the object, the generated subgoals from SuSIE precisely guide the low-level controller, improving low-level skill execution for difficult manipulation tasks.

| | Task | Oracle GCBC | SuSIE (Ours) |
|---|---|---|---|
| | Eggplant on plate | 0.7 | **1.0** |
| | Carrot on plate | 0.8 | **0.9** |
| Scene A | Eggplant in pot | **1.0** | 0.7 |
| | Average | 0.83 | **0.87** |
| | Bell pepper in pot | 0.1 | **0.5** |
| Scene B | Bell pepper in bowl | 0.0 | **0.5** |
| | Average | 0.05 | **0.50** |
| | Toothpaste in bowl | **0.7** | 0.6 |
| | Crayon in bowl | **1.0** | **1.0** |
| Scene C | Spoon in bowl | 0.8 | **0.9** |
| | Bowl to top | 0.6 | **1.0** |
| | Average | 0.78 | **0.88** |
| | Move slider right/left | 0.52 | **0.86** |
| | Turn on/off light bulb | 0.74 | **0.96** |
| CALVIN | Open/close drawer | **1.00** | 0.98 |
| | Turn on/off LED | 0.36 | **1.00** |
| | Average | 0.66 | **0.95** |

**Table 3: (Comparison to GCBC with oracle goals)** Executing generated subgoals improves performance even compared to executing privileged ground-truth final goals.

## 5.3 DOES SuSIE IMPROVE PRECISION AND LOW-LEVEL SKILL EXECUTION?

Our real-world and simulated results clearly demonstrate the efficacy of SuSIE in executing novel language commands in a variety of scenes. In Section 5.2, we hypothesize that the advantage of SuSIE is twofold: semantic generalization due to Internet and video pretraining, and low-level precision due to subgoal guidance. In this section, we aim to validate the latter half of this hypothesis. To this end, we train an **Oracle GCBC** policy, which is identical to the low-level policy in SuSIE except that it is trained with a goal horizon that extends to the end of each trajectory. At test time, we provide the policy with a privileged *real goal image* of the fully completed task, eliding the need for

| | Task | BridgeData Only | BridgeData + Something-Something |
|---|---|---|---|
| Scene B | Bell pepper in pot | 0.2 | **0.5** |
| | Bell pepper in bowl | 0.4 | **0.5** |
| | Average | 0.30 | **0.50** |
| Scene C | Toothpaste in bowl | **0.7** | 0.6 |
| | Crayon in bowl | 0.9 | **1.0** |
| | Spoon in bowl | **0.9** | **0.9** |
| | Bowl to top | 0.7 | **1.0** |
| | Average | 0.80 | **0.88** |

Table 4: (**Comparison to BridgeData-only subgoal generation model**) Cotraining the subgoal generation model on the Something-Something dataset improves performance in Scenes B and C, which are unseen in BridgeData V2.

any semantic understanding. Thus, the only advantage of SuSIE over Oracle GCBC is the subgoal guidance, while SuSIE is disadvantaged by additionally needing to interpret the language instruction and generate subgoals for the correct task.

As demonstrated in Table 3, although Oracle GCBC is a strong baseline, SuSIE is still the best-performing method on average. In particular, we observed that Oracle GCBC is still unable to grasp the bell pepper in Scene B due to a lack of low-level precision. Figure 3 provides a qualitative visualization of SuSIE grasping the bell pepper while Oracle GCBC and RT-2-X fail to do so. These results validate the hypothesis that, in addition to enabling semantic generalization, SuSIE also improves low-level precision and dexterity by decomposing the problem into a two-level hierarchy.

### 5.4 Are Internet and Video Training Crucial for Zero-Shot Generalization?

Finally, we conduct an experiment to understand the effect of both Internet-scale pretraining (in the form of InstructPix2Pix initialization) and video cotraining (in the form of training on BridgeData V2 and Something-Something simultaneously). We train two additional subgoal generation models: one robot-only model trained with InstructPix2Pix initialization but only on BridgeData V2, and one from-scratch model trained without InstructPix2Pix initialization at all. The from-scratch model uses the same frozen text encoder, image encoder, and UNet architecture as InstructPix2Pix, but initializes the UNet with random weights.

We present sample generations from these models in Figure 4. The generations of the from-scratch model are consistently low-quality, either misunderstanding the task, producing significant artifacts, or failing to edit the image at all; this demonstrates the importance of initializing from Internet-scale pretrained weights. As one might expect, the BridgeData-only model is on par with the full Something-Something model on in-distribution tasks, which come from the BridgeData V2 validation set. However, on out-of-distribution tasks, the BridgeData-only model has an increased propensity to hallucinate background items or misunderstand the task. We found that for moving the wooden bowl, in particular, the full Something-Something model is the only one that can produce fully correct subgoals.

As additional evidence, we present a quantitative evaluation of the BridgeData-only model on the tasks from Scenes B and C in Table 4. In real-world task execution, the difference between the BridgeData-only model and the full Something-Something model is not as pronounced as in the qualitative samples. This is because we find that the low-level goal-conditioned policy is fairly robust to hallucinations of background items, suggesting that it primarily attends to the gripper pose and the target object. However, the overall increased quality of the subgoals from the full model still produces a small quantitative improvement on average.

## 6 Discussion and Future Work

We presented a method for robotic control from language instructions that generates subgoals to guide a low-level goal-conditioned policy. The subgoals are generated by an image-editing diffusion model finetuned on video data. This system improves both zero-shot generalization to new objects and the overall precision of the policy because the subgoal model incorporates semantic benefits from pretraining and commands the low-level policy with fine-grained guidance. Our experiments show that SuSIE improves over prior techniques on the CALVIN benchmark and improves visual

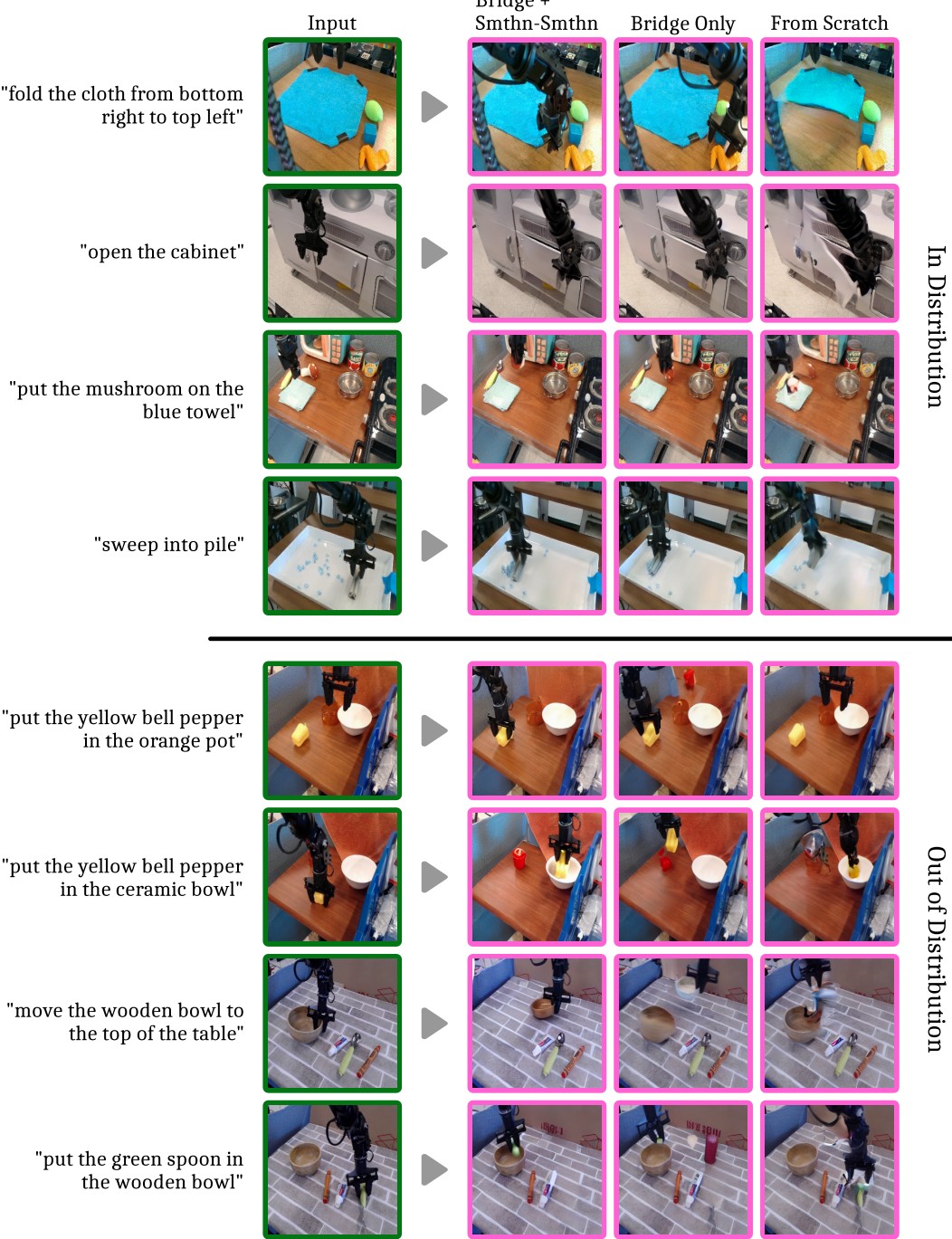

**Figure 4: (Comparison of subgoal quality)** A comparison between 3 subgoal generation models: one trained on both robot and video data (Bridge + Smthn-Smthn), used throughout this paper; one trained on robot data only (Bridge Only); and one trained without InstructPix2Pix initialization (From Scratch). The top half (In Distribution) comes from the BridgeData V2 validation set, while the bottom half (Out of Distribution) comes from our evaluation Scenes B and C, unseen in BridgeData V2. Both Internet-scale pretraining and video cotraining and important for generating high-quality subgoals, especially in the zero-shot generalization setting.

generalization and low-level control on real-world manipulation tasks. In the real world, SuSIE outperforms language-conditioned behavioral cloning, a privileged goal-conditioned policy that gets access to a ground-truth final goal, as well the state-of-the-art instruction-following approach, RT-2-X, which is trained on more than an order of magnitude more robot data (over 1.1 million trajectories for RT-2-X, vs. 60k for ours).

Our method is simple and provides good performance, but it does have limitations that suggest promising directions for future work. For instance, the diffusion model and the low-level policy are trained separately, meaning the diffusion model is unaware of the capabilities of the low-level policy — it is trained on the same data, but assumes that anything that is reachable in the data can also be reached by the policy. We hypothesize that performance can be improved by making the diffusion model aware of the low-level policy's capabilities. More broadly, we found the performance of our method to often be bottlenecked by the performance of the low-level policy, suggesting that addressing either of these limitations might lead to a more performant method for importing Internet-scale knowledge into robotic manipulation.

## ACKNOWLEDGMENTS

We thank Quan Vuong, Vincent Vanhoucke, Karl Pertsch, and Oleg Rybkin for their feedback on earlier versions of the paper. We thank Anurag Ajay, Zoey Chen, and Quan Vuong for their support with baseline approaches. Aviral Kumar thanks Abhishek Gupta, Sherry Yang, and Boyi Li for informative discussions. We also thank the TRC program from Google Cloud for their cloud TPU donations that were crucial for running our experiments.

This research was partly supported by the Office of Naval Research through N00014-22-1-2773 and N00014-20-1-2383, AFOSR FA9550-22-1-0273, and NSF grant IIS-2150826. Kevin Black and Pranav Atreya are supported by the NSF Graduate Research Fellowship. Mitsuhiko Nakamoto is supported by the Nakajima Foundation Fellowship.

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

| | CALVIN | Smthn-Smthn | Bridge |
|---|---|---|---|
| $k_{\min}$ (Eq. 1) | 20 | 11 | 11 |
| $k_{\max}$ (Eq. 1) | 22 | 14 | 14 |
| $k_\delta$ (Eq. 2) | 2 | N/A | 6 |

| | CALVIN | Real-world |
|---|---|---|
| $k_{\text{test}}$ (Alg. 1) | 20 | 20 |

Table 5: **(Goal sampling hyperparameters)** The first two hyperparameters, $k_{\min}$ and $k_{\max}$, are used for training the image-editing model. $k_\delta$ is used for training the low-level policy. All 3 of these correspond to a particular dataset. $k_{\text{test}}$ is a test-time hyperparameter that corresponds to an evaluation setup.

## A  IMPLEMENTATION DETAILS

We provide implementation details for SuSIE and the baselines. Table 5 documents goal sampling hyperparameters.

### A.1  SuSIE IMPLEMENTATION DETAILS

#### A.1.1  IMAGE-EDITING DIFFUSION MODEL

We finetune InstructPix2Pix [9] using similar hyperparameters to the initial InstructPix2Pix training. We use the AdamW optimizer [40] with a learning rate of 1e-4, a linear warmup of 800 steps, and weight decay of 0.01. We track an exponential moving average (EMA) of the model parameters with a decay rate of 0.999 and use the EMA parameters at test time. We train for 40k steps with a batch size of 1024 on a single v4-64 TPU pod, which takes 17 hours.

For simulation, we train the model exclusively on the CALVIN dataset. For real-world experiments, we cotrain with a sampling mixture of 60% Something-Something and 40% BridgeData V2.

At test time, we use an image guidance weight of 2.5 and a text guidance weight of 7.5. We use the DDIM sampler [56] with 50 sampling steps.

#### A.1.2  GOAL-REACHING POLICY

We use a diffusion model for our goal-reaching policy since recent work has shown that diffusion-based policies can better capture multi-modality in robot data [12, 23], leading to improved performance across a variety of tasks. In our implementation (which follows Walke et al. [59]), the observation and goal image are stacked channel-wise before being passed into a ResNet-50 image encoder. This image encoding is used to condition a diffusion process that models the action distribution. The diffusion process uses an MLP with 3 256-unit layers and residual connections. Following Chi et al. [12], rather than predicting a single action, we predict a sequence of 4 actions to encourage temporal consistency. We use the Adam optimizer [36] with a learning rate of 3e-4 and a linear warmup of 2000 steps. We train with a batch size of 256 for 445k steps on a single v4-8 TPU VM, which takes 15 hours. We augment the observation and goal with random crops, random resizing, and color jitter. For goal sampling, we use $k_\delta = 6$, meaning we sample goals uniformly from the range $[0, 20]$, since the corresponding $k_{\max}$ used with BridgeData V2 is 14 (see Table 5).

At test time, we have several options for how to predict and execute action sequences. Chi et al. [12] use receding horizon control, sampling $k$-length action sequences and only executing some of the actions before sampling a new sequence. This strategy can make the policy more reactive. However, we found that the robot behavior was quite jerky as the policy switched between different modes in the action distribution with each sample. Instead, we use a temporal ensembling strategy similar to Zhao et al. [65]. We predict a new 4-action sequence at each timestep and execute the dimension-wise mean of the last 4 predictions for that timestep.

## A.2 Baseline implementation details

### A.2.1 MOO

MOO [57] utilizes a mask to represent the target objects and incorporates it as an additional channel in the observation. Specifically, they train a language-conditioned policy that takes a 4-channel image and a language command as inputs. To acquire the mask for target objects, the OWL-ViT [46] object detector is employed. This detector is an open-vocabulary object detection model, pretrained on Internet-scale datasets, and it is used to extract the bounding boxes of the objects of interest from the image. For tasks like "move X to Y," MOO extracts the bounding box for X, representing the object of interest, and Y, indicating the target place. A mask is then created where the pixel at the center of the predicted bounding box is assigned a value of 1.0 for X and 0.5 for Y.

**Extracting object entities from BridgeData V2 language annotations.** In order to obtain the mask, it is necessary to extract the entities corresponding to the object of interest, denoted as X, and the target place, Y, from the language command. In the MOO paper, the authors assume that the language in their dataset is structured in a way that facilitates easy separation of X and Y. Specifically, they employ a dataset that exclusively consists of language annotations such as "pick X," "move X near Y," "knock X over," "place X upright," and "place X into Y."

Given that the language annotations in BridgeData V2 are diverse and unstructured, it is challenging to naively extract X and Y. We utilized OpenAI's gpt-3.5-turbo-instruct model [5] to extract the object of interest and the target place (if any) from the language annotation and input them into OWL-ViT to create masks. It is worth noting that due to the unstructured nature of BridgeData V2, the extracted bounding boxes in training data are sometimes incorrect as shown in Figure 5. We then trained a mask-conditioned LCBC policy using the same architecture as described in Appendix A.2.3. Following MOO, we removed X and Y from the prompt and replaced the entity X with "object of interest" and the entity Y with "target place". For example, given the language prompt "put the eggplant in the pot", we use a modified prompt "put object of interest in target place" as the input to the policy during both train and test time.

**Test time.** During test time, we manually give ground-truth masks to the policy. To enable this, we build a simple interface on the robot machine, allowing the evaluator to create the masks by clicking on the initial camera image at the beginning of each trial.

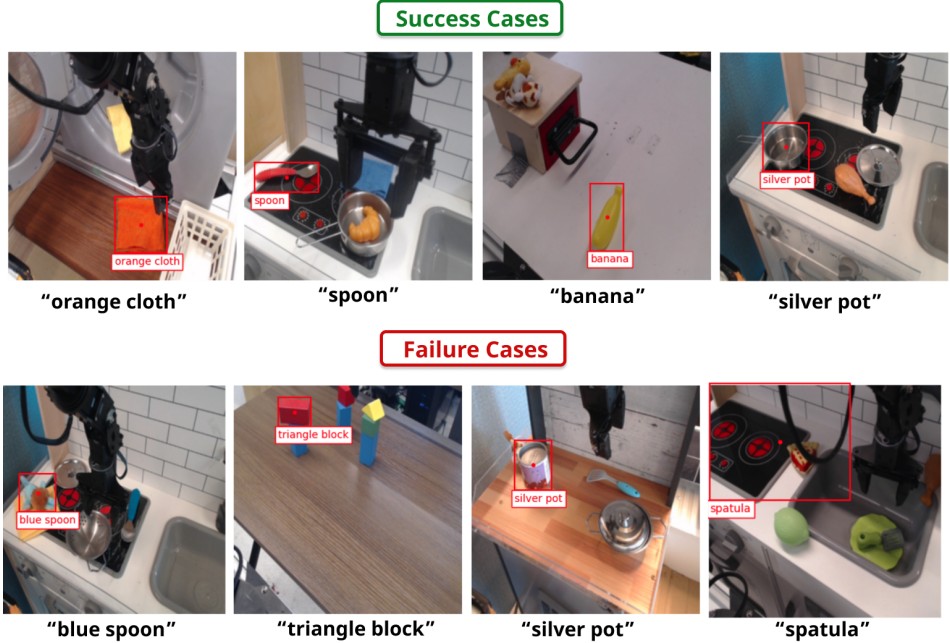

**Figure 5: (Examples of extracted bounding boxes for MOO).** Despite our best attempts, the bounding boxes extracted at train time are noisy due to the unstructured nature of BridgeData V2. This might have led to MOO's poor performance in our evaluations.

### A.2.2 UNIPI

UniPi [16] trains a video diffusion model $p_\theta(\tau \mid \mathbf{s}_0, l)$ to generate a sequence of frames given a language command and an initial frame. The original paper employs the model architecture from Imagen Video [26, 27]. To achieve higher resolution and longer videos for their real-world results, the authors leverage a 1.7B 3D UNet and four pretrained super-resolution models from Imagen Video, with 1.7B, 1.7B, 1.4B, and 1.2B parameters, respectively. Since the original models and code are not publicly available, we tried to replicate their approach in two different ways.

**UniPi (HiP, Ajay et al. [2])** For the second approach, we followed the UniPi replication in Ajay et al. [2]. We trained a latent video diffusion model from PVDM [62], building upon the codebase https://github.com/sihyun-yu/PVDM where we added first-frame conditioning. We first trained the video autoencoder to project a video of size $16 \times 128 \times 128$ into a latent space, followed by training a PVDM-L model that uses a 2D UNet architecture. We used a Flan-T5-Base [13] encoder to obtain the language embeddings.

**UniPi (Ours).** We implemented a 3D UNet video diffusion model, following Ho et al. [27, 26] and adding UniPi's first-frame conditioning. Due to limited compute, we did not train spatial/temporal super-resolution models; instead, we trained a 3D UNet-based diffusion model to directly generate images with a resolution of $128 \times 128$. The model includes 4 residual blocks, with (input channels, output channels) as follows: $(64, 64)$, $(64, 128)$, $(128, 256)$, and $(256, 640)$. The model is trained to produce the trajectory with a fixed horizon of 10 frames $\tau_t = \{s_t, s_{t+1}, \ldots, s_{t+9}\}$ conditioned on the current frame $s_t$ and language command. We used a frozen pretrained CLIP [52] encoder to obtain the language embeddings.

**Data and training details.** Following the UniPi baseline from HiP [2], we utilized Ego4D [20] to instill the model with Internet-scale knowledge. For UniPi (Ours), we first pretrained the video model on Ego4D for 270K steps, and finetuned it on the robotics dataset (CALVIN for simulation and BridgeData V2 for the real world) for an additional 200K steps. We used a batch size of 4 during training. For UniPi (HiP), we jointly trained a single model on Ego4D, BridgeData V2, and CALVIN at the same time. The autoencoder was trained for 85K steps, and the PVDM-L model was trained for 200K steps. We used a batch size of 8 during training.

**Inverse model and test time control.** To extract actions from generated videos, we trained a Gaussian inverse dynamics model $\pi_\phi(\mathbf{a} \mid \mathbf{s}_t, \mathbf{s}_{t+1})$ to predict the action from two adjacent frames. The model consists of a ResNet-50 image encoder and 3 256-unit MLP layers. During test time, given the current observation $s_t$ and the language command $l$, we synthesize $H$ image frames from the video model and apply the inverse dynamics model to obtain the corresponding $H - 1$ actions. The predicted actions are executed, and we generate a new video from $s_{t+H-1}$ and repeat the process.

**Generated videos.** While the quality of the video model trained on the simulation dataset is good enough for solving the tasks on the CALVIN benchmark as shown in Table 1, we found that it is nontrivial to obtain high-quality generations for the real-world scenes. We show examples of generations in Appendix C.

### A.2.3 LANGUAGE-CONDITIONED BEHAVIOR CLONING (LCBC)

For the LCBC baseline, we use the same architecture and hyperparameters as the low-level policy in SuSIE. We encode the language instruction using MUSE [61] and feed it into the ResNet-50 image encoder using FiLM conditioning [51]. We found that this policy outperformed the LCBC policy used in Walke et al. [59] and Myers et al. [47] (which was not a diffusion policy), as well as the RT-1 architecture [6] trained on BridgeData V2 (also appearing in Walke et al. [59]).

## B EXPERIMENTAL SETUP

**CALVIN.** We follow the evaluation setup from Mees et al. [45]. During each trial, the policy gets 360 timesteps total to complete a chain of 5 instructions, only moving on to the next instruction once the previous one is completed. Subsequent instructions in each language chain are chosen by the simulator based on environment affordances after completing the previous language instruction. Results in Table 1 are averaged over 100 trajectories, enabling direct comparison with results from Ge et al. [18], which are also averaged over 100 trajectories. Results for UniPi (HiP) and UniPi (Ours)

were averaged over 25 trajectories, due to the computational cost associated with querying the video diffusion models, and the relatively short video generation horizon (10 timesteps) requiring frequent regeneration.

To obtain comparisons with an oracle GCBC policy in Table 3, we sampled ground truth goal image states from the simulator to feed to the oracle policy by manually resetting the environment internal state to the state corresponding to the goal being achieved. This manual resetting process was only amenable for certain tasks in CALVIN, namely the ones in Table 3. Results in the table correspond to a total of 200 evaluation trajectories.

**Real world.** Figure 2 provides an exhaustive list of all 9 tasks we evaluate in the real world. We conduct 10 trials per task and report the average success rates. We vary the positions and orientations of various objects between trials, but use the same set of positions and orientations for evaluating all methods. In Scene A, we change the positions and orientations of the plate, eggplant, and carrot every trial; the pot always remains in the drying rack. For "put carrot on plate", we start the carrot in the pot for 4 out of 10 trials, and in the sink otherwise. The eggplant always starts in the sink. In Scene B, we change the positions of the bell pepper, orange pot, and ceramic bowl every trial, performing 5 trials where the target container is closer to the target object and 5 trials where it is further away. In Scene C, for "put X in the wooden bowl," we change the positions of the 3 objects every trial while the wooden bowl remains on the left; we also randomize the rotations of the objects up to 45 degrees from vertical. For "move the wooden bowl to the top of the table," we change the horizontal position of the bowl every trial, but it always starts at the bottom of the table.

## C   QUALTITATIVE EXAMPLES OF VIDEO GENERATION

Figure 6 shows the video frames generated by the UniPi (HiP) model. The model fails to generate the robot's gripper accurately and consistently produces blurry results. For this reason, we did not evaluate this model on the real robot. Instead, we evaluated UniPi (Ours) model for the real-world tasks. Figure 7 shows examples of success and failure cases of the test trials. While the robot succeeded in a few trials, we observed that in most cases the generated videos suffered from hallucination and physical inconsistency, which led to UniPi's poor performance.

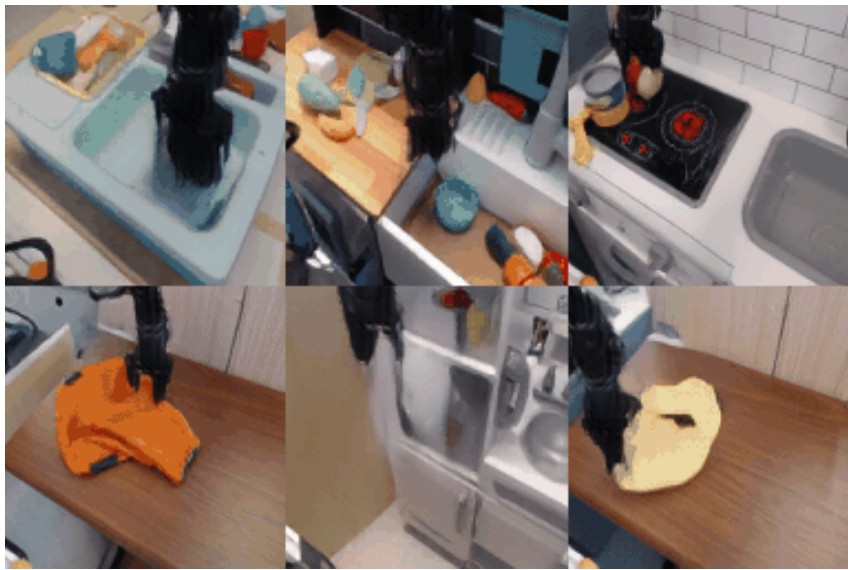

**Figure 6: (UniPi (HiP) example generations)**. Observe that the model is not able to generate a clear robot gripper, and produces videos that are blurry and visually poor.

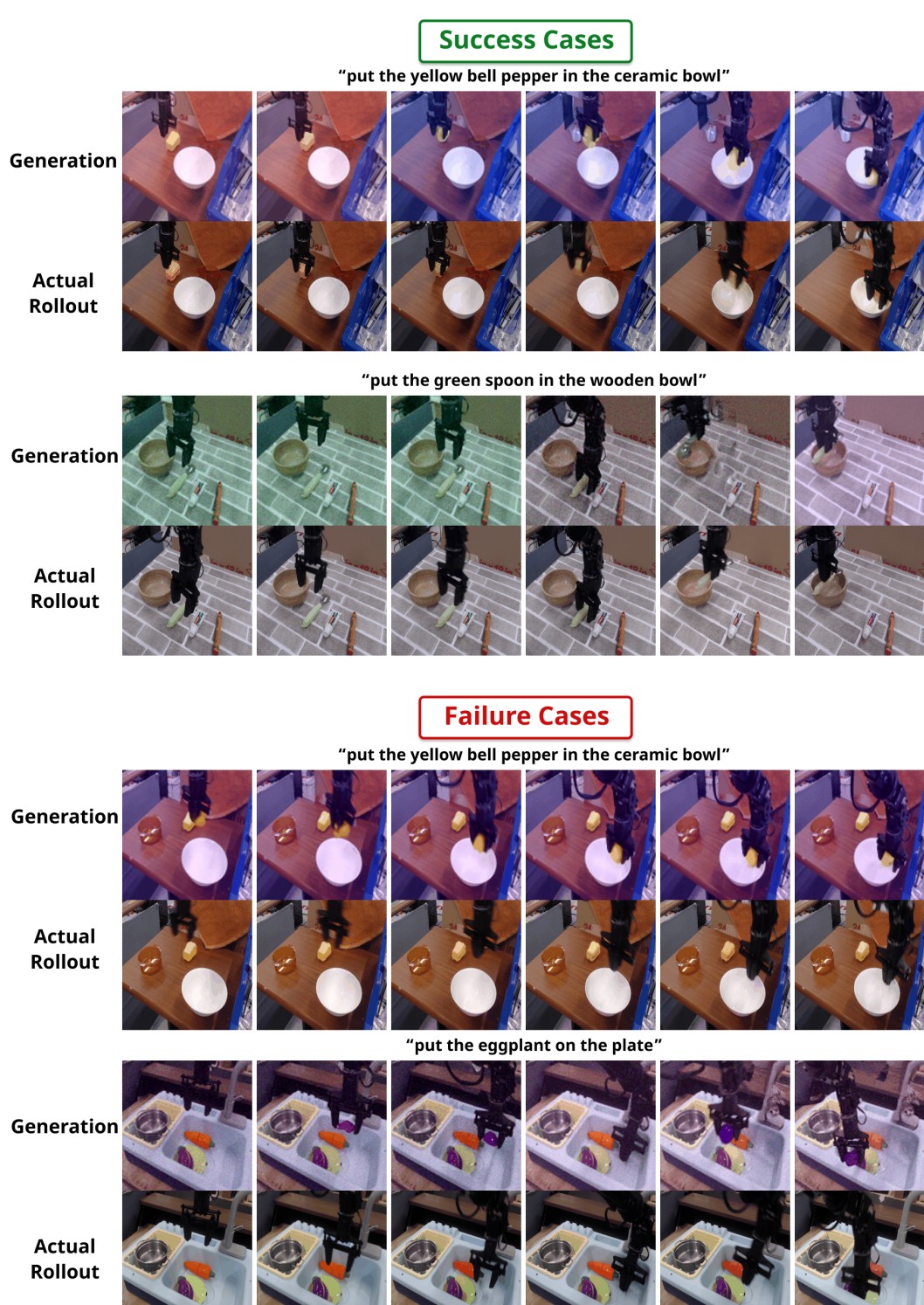

**Figure 7:** (**UniPi (Ours) example generations and the resulting robot behaviors**). While the robot succeeded in a few trials, we observed that in most cases the generated videos suffered from hallucination and physical inconsistency, which confused the low-level policy.

# D QUALITATIVE EXAMPLES OF SuSIE ON CALVIN

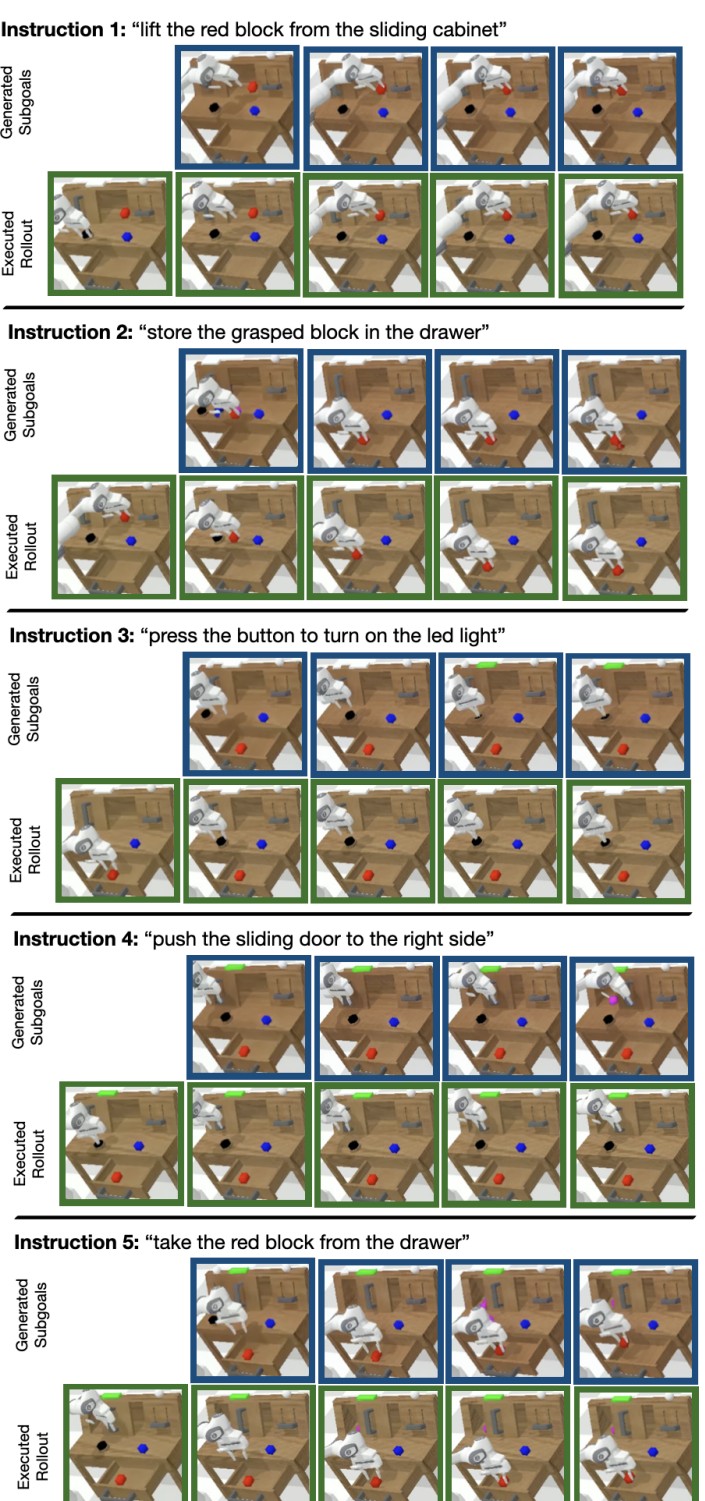

**Figure 8: (Example rollouts in CALVIN.)** Visualization of SuSIE-generated subgoals and trajectory rollouts on a successful 5-instruction language chain in CALVIN.

