# OpenReview forum: "Zero-Shot Robotic Manipulation with Pre-Trained Image-Editing Diffusion Models"
_ICLR.cc/2024/Conference — ICLR 2024 poster_

### Official Review · Reviewer_choL · 2023-10-29

**Soundness:** 3 good
**Presentation:** 3 good
**Contribution:** 3 good
**Rating:** 6
**Confidence:** 4

**Summary:**

This paper introduces the interesting approach to utilize the Internet-scale semantic knowledge for solving robotic tasks by leveraging pre-trained image-editing models. The main components consist of two part: 1) Image-editing model that generates future subgoals given the current state and task description in language. This model is first pre-trained with Internet-scale data and fine-tuned with in-domain robotic data. This guides low-level policy by specifying near future subgoals. 2) Low-level goal-conditioned policy that accomplishes generated subgoals. The method is evaluated on simulated (CALVIN) and real-world environments, where the proposed method shows better performance in generalization ability that deals with solving novel tasks, objects and environments.

**Strengths:**

- The paper starts with the interesting questions such as "how to effectively inject semantic knowledge into robot control problems?" and "how to better utilize the Internet-scale foundation model to solve the problem of robot learning?", which are quite important problems these days.
- This paper is well-written and easy to follow. The idea is very simple, yet consistently perform well in overall experiments and shows effectiveness.
- The experimental settings are well designed for the purpose the authors want to show such as zero-shot generalization to novel objects and environments, so the experimental results seem to show effectively that the proposed method is better.

**Weaknesses:**

- I think the analysis of the results of the real-world experiment is a little insufficient. In the results shown in Table 2, RT-2-X performs much better on scene C, so why does it perform better only on scene C? RT-2-X is said to have been learned through larger model and more data, but that does not show such performance in scene A and B, but analysis on this part seems to be insufficient.
- In order to generate subgoal well, an overall temporal understanding of the task is required. But there is not much clear analysis or intuition why the fine-tuned image-editing model performs better than the video prediction model.
- The results from the simulated environment(CALVIN) don’t show much significant performance difference from other language-conditioned policy baselines.

**Questions:**

- It would be better if the authors could provide additional experiments with the baselines which utilize low-dimensional image embeddings as subgoals rather than high-dimensional raw images to show that image-editing model is effective as a subgoal generator.
- When learning the goal-reaching policy, the model is learned to provide subgoals within k steps of any state. But I wonder if how much the change in k value makes a difference in performance. It would be nice to see additional experiments showing changes in performance according to the value of k.

---

> ### Author Response · Authors · 2023-11-19
>
> Thank you for your feedback and a positive assessment of our work. We address your concerns below and have updated the paper to clarify each of the questions.
>
> **Please let us know if your questions are resolved, and if so, we would be grateful if you are willing to upgrade your score. We are happy to discuss further if any questions are remaining.**
> ___
>
> > **RT-2-X performs much better on scene C, so why does it perform better only on scene C?**
>
> After the submission deadline, we made some improvements to the subgoal diffusion model as we discuss in Appendix A.4. We re-evaluated in all scenes and found that our method now outperforms RT-2-X, even in Scene C. You can find the updated results at [this link](https://ibb.co/hKDHFFB), as well as in Table 2 in the updated PDF.
>
> Regarding why RT-2-X performs badly in Scene B, this is because each scene is challenging in different aspects. **Semantically**, Scene C is the most challenging since the robot needs to carefully ground the language command to identify the unseen objects and requires a high generalization capability. However, each object is in a shape that is relatively easy to grasp. In contrast, Scene B is challenging in the sense of manipulating the plastic bell pepper --- this object requires particular precision to grasp due to being light, slippery, and almost as wide as the gripper. We have updated the PDF to clarify this point.
>
> We found that while RT-2-X is good at semantic generalization for unseen objects, it lacks precision in picking up challenging objects like the bell pepper. In contrast, as we discussed in Section 5.4, the subgoal generated by SuSIE improves the precision of low-level control for manipulating these challenging objects.
> ___
>
> > **There is not much clear analysis or intuition why the fine-tuned image-editing model performs better than the video prediction model.**
>
> Modeling an entire video puts a very high burden on the generative model, requiring the frames to obey strict physical consistency. Unfortunately, we find that current video models often produce temporally inconsistent frames (hallucinations), which only confuse the low-level controller, inhibiting it from completing the task. After the submission deadline, we evaluated the video model baseline (UniPi) on the real robot but we found that it performed poorly due to the hallucinations. We show the updated table at [this link](https://ibb.co/hKDHFFB). Qualitative examples of UniPi rollouts can be found at [this link](https://ibb.co/MpBtjSJ). While it’s definitely possible that better video models in the future can reach better performance, we found no open-source video model that could get good results despite considerable effort (no code for the original UniPi work was ever released, though we tried to follow a reimplementation used by one of its authors for [another paper](https://arxiv.org/abs/2309.08587)).
> ___
>
> > **The results from the simulated environment(CALVIN) don’t show much significant performance difference.**
>
> Perhaps there is some misunderstanding here. The simulation results of SuSIE in the CALVIN benchmark (0.75, 0.46, 0.19, 0.11, 0.07) improve over the next best baseline AugLC, which scores (0.69, 0.43, 0.22, 0.09, 0.05), by a fair margin.
>
> Furthermore, after the submission deadline, we realized that we used sub-optimal values of classifier-free guidance parameters for the SuSIE’s evaluations on CALVIN. After simply tuning these values, it led to further performance gains posted in the latest draft of the paper **(0.87, 0.69, 0.49, 0.38, 0.26)**. Kindly refer to these numbers as the measured performance of SuSIE. The updated table can be seen at [this link](https://ibb.co/bJnJcPL).
> ___
>
> > **It would be better if the authors could provide additional experiments with the baselines which utilize low-dimensional image embeddings as subgoals.**
>
> We are willing to add this baseline, but we are not sure which method this is referring to. Do you have any specific methods or papers in mind that you would suggest we include?
> ___
>
> > **I wonder if how much the change in k value makes a difference in performance.**
>
> While a full ablation of this hyperparameter would be ideal, retraining the diffusion model and re-evaluating the policy many times is prohibitively expensive. As such, we chose $k$ by inspecting each dataset and choosing a value that will be about one-third of the average length of the trajectories in that dataset. For example, in the case of BridgeData, since most of the trajectories are around 20-40 steps, we set $k$ to 10. In the case of the CALVIN dataset, as most of the demos are 65 steps, we set $k$ to 20. Though simple, we found this choice to generally lead to good results. We have added this clarification in Appendix A.6 in the updated PDF.

---

> ### Author Response · Authors · 2023-11-21
> **A friendly reminder**
>
> Dear Reviewer choL,
>
> Thank you for your feedback! We were wondering if you have gotten a chance to go over our responses. We would appreciate it if you could have a look and let us know if your questions are resolved. We are happy to discuss further.
>
> Thanks!

---

> > ### Comment · Reviewer_choL · 2023-11-21
> > **Official Comment by Reviewer choL**
> >
> > Thank the authors for your kind explanations. Thanks to your explanations and the newly updated results, most of the concerns I had have been addressed. Also your clarifications helped me understand better about the contributions of the paper. But I have one more question: I understand the reason for the improved performance in the new CALVIN experiment evaluation, but could you provide a more specific explanation of the changes that led to SUSIE's better performance in real-world experiments using BridgeData?

---

> ### Author Response · Authors · 2023-11-22
>
> Hi, thanks for your response and the positive feedback! We are glad to hear that most of the concerns have been addressed!
> ___
>
> > **a more specific explanation of the changes that led to SUSIE's better performance in real-world experiments**
>
> Thanks for your question! We have updated the paper to discuss the improvements we made to SuSIE since the submission in detail in Appendix A.4. Concretely, the main source of improvement is an updated subgoal diffusion model. Since the deadline, we trained a version of our subgoal diffusion model by fine-tuning on a combination of the Bridge dataset and a human video dataset, [Something-Something](https://developer.qualcomm.com/software/ai-datasets/something-something). Note that human video datasets can be incorporated seamlessly into our approach, as the diffusion model does not make any specific use of robot embodiment or actions and only requires a tuples of frames and future frames, annotated with a natural language command. Upon visual inspection, we found that the diffusion model trained on a combination of Bridge data + human video data produced subgoals with higher quality than the one trained on robot data only (see [this figure](https://imgbb.com/4fR6NQk)). Upon evaluating the full SuSIE approach with this model, we found that we were able to outperform RT-2-X in Scene C, while preserving or improving performance of SuSIE across the other two Scenes. This is despite the fact that RT-2-X utilizes a 50 times bigger model than SuSIE (55B vs 1B), and is trained on orders of magnitude more data compared to Bridge + Something-Something, which also includes 1 billion pairs of open-domain vision-language data. Due to this reason, we believe that this updated comparison is more fair than the comparison between SuSIE and RT-2-X in the original submission since RT-2-X co-trains on open-domain vision-language data alongside robot data, whereas the SuSIE model in the submission was not fine-tuned on any open domain data (but the new model utilizes some open-domain data, though in a substantially lesser quantity).
>
> Overall, we believe that this result highlights a strength of our approach, demonstrating that the subgoal diffusion model can leverage knowledge from not only robot data but also from open-domain video datasets, that are unrelated to the robot. Due to an uncaught wording error, our previous response to your comment seemed to imply that the improvements came from a better low-level policy, but this was incorrect (we apologize that this wording error was not caught earlier) -- the improvements came from a better diffusion model, and we have edited our response above to correct this wording error.
> ___
>
> We hope this resolves your remaining concern! Thank you!

---

> > ### Author Response · Authors · 2023-11-23
> >
> > Hi, thanks again for your positive feedback and the meaningful discussion. As the author-reviewer discussion period draws to the close in around 10 hours, we were hoping to check in with you if you have gotten a chance to go over our latest responses and if your remaining concerns are addressed. **If so, we would be grateful if you are willing to upgrade your score. We are happy to discuss further! Thanks!**

---

> > > ### Comment · Reviewer_choL · 2023-11-23
> > > **Official Comment by Reviewer choL**
> > >
> > > I appreciate the authors' efforts in addressing my concerns and providing additional experimental results to enhance the paper. But, I still see this paper similar with what I had previously thought in overall. So I will keep my score same. While I still hold the same overall impression as before, I want to express my gratitude for your efforts.

---

### Official Review · Reviewer_rx11 · 2023-10-30

**Soundness:** 3 good
**Presentation:** 3 good
**Contribution:** 2 fair
**Rating:** 6
**Confidence:** 3

**Summary:**

The paper discusses the challenge of enabling robots to recognize and respond to novel objects and situations that were not part of their training data. While robotic manipulation datasets have expanded, they can never encompass every potential object or context a robot might encounter. To address this, the paper proposes integrating semantic knowledge into language-guided robotic control by leveraging pre-trained vision and language models.

In essence, the proposed approach, SuSIE, decomposes the robotic control problem into two phases. In the first phase, an image-editing model is fine-tuned on robot data, allowing it to generate hypothetical future subgoals based on the robot's current observation and a natural language command. These subgoals serve as intermediate objectives that guide the robot towards accomplishing the user-specified task. Importantly, the image-editing model doesn't need to understand how to achieve these subgoals; it only generates them based on semantics. In the second phase, a low-level goal-reaching policy is employed to help the robot reach these hypothetical subgoals. This policy focuses on inferring visuo-motor relationships for correct actuation but doesn't require a deep understanding of semantics. By breaking down the task into smaller, semantically guided subgoals, the overall problem becomes more manageable. SuSIE is compared against a diverse set of approaches, and shown to achieve comparable or sometimes better performance.

**Strengths:**

- Using VLMs to synthesize subgoals is a novel approach (however, it's unclear how these subgoals are trained - see questions).
- Decoupling the subgoal generation and the low-level control policy learning components seems to be effective (but authors seem to have contradicting opinions, see my comments in the questions part).
- Experiment setup was sound (different scenes with seen/unseen objects, with increasing difficulty by involving more unseen objects and other clutter)
- A good amount of comparative analysis is presented. The authors present both simulation and real-life experiments. Furthermore, they have shown instances where their models perform well, but they have also demonstrated cases where their models perform worse than previous models and provided valid explanations for these outcomes.

**Weaknesses:**

- Subgoal training procedure is unclear - see questions.
- Some parts could be reformulated/reworded better in Section 4 to reduce ambiguities and improve reproducibility.
  - In 4.1, “k” is used for two different contexts (hyperparameter and index).
  - In 4.1, “q” distribution appears to be a uniform discrete distribution in [i, i + k], but it remains undefined.
- The information regarding the datasets used can be provided more comprehensively and at an earlier stage in the paper.
- Some results are missing and/or unclear - see questions.

**Questions:**

-	The figures depict a properly selected sub-goals for the scenes, such as a pre-grasp pose in figure 1. However, it is unclear if or how the sub-goal synthesis procedure is optimized to output such intuitive sub-goals, rather than less important/effective transitionary scenes that simply happen to follow the input pose. In essence, how do you select/optimize for the feasible/relevant subgoals?
- Table 1 uses some success rates from another paper (Ge et al., 2023). However, the approach proposed in this cited paper achieves higher success rates (see 'Ours' row in Table 2 of Ge et al.) than the ones reported on this paper. Why did you omit this result? Ge et al. seem to also address out-of-distribution generalization. Other than 1-step chain case (theirs 72% vs. SuSIE 75%), the longer chains achieve better results than SuSIE.
-	It was mentioned that Scene C should be a more difficult task than Scene B, however the results in 5.3 show that both MOO, RT-2-X and SuSIE perform significantly better on scene C than on B. Can you explain what the reason might be? Is it related to the VLM or the low-level control policy?
-	The model is illustrated to depict sub-goals nicely as it is able to distinguish between particular objects to orient the gripper towards. But there are non-visual properties of these objects that low-level controllers need to consider to perform pick and place tasks, such as weight and friction. This model however seems to rely on visual properties only. Would the robot in this pipeline be able to pick up such challenging objects, such as a heavy and slippery object, even after a few tries?
-	InstructPix2Pix is not capable of performing viewpoint changes, how would this challenge be reflected in a real-life or research scenario where there are often multiple or moving cameras present?
-	Often, the gripper may be obscured in the camera view due to other objects, such as the rest of the robot. The first phase may output these subgoals as a suggestion of pre-grasp or grasp pose. However, due to decoupling, the low-level controller would have no way of knowing this is the case. In the image of the scene, any pose of the gripper would look the same. How would the scene play out, would the robot be able to recover or get stuck?
-	Can you elaborate on the role/effect of the conditional policy (Section 4.2)? It seems to enforce the robot to follow the trajectory sampled in the dataset when it encounters a similar state and subgoal. What happens when it encounters, say, a similar current state but a different subgoal (that was not in the dataset) during test time?
-	Does your robot dataset consist of only completely successful trajectories? If not, could a subgoal be generated for failure (e.g., robot holds the desired object in the current state, but subgoal generates a state where it is dropped? If so, how do you handle such cases?
-	There is a minor error in the table references. In Section 5.3 and A2.2.2, you mentioned "Table 5.3," which refers to "Table 1."

---

> ### Author Response · Authors · 2023-11-19
> **Comment by Authors (Part 1)**
>
> Thank you for your feedback and a positive assessment of our work. We address your concerns below and have updated the paper to clarify each of the questions.
>
> **Please let us know if your questions are resolved, and if so, we would be grateful if you are willing to upgrade your score. We are happy to discuss further if any questions are remaining.**
> ___
>
>
> > **In essence, how do you select/optimize for the feasible/relevant subgoals?**
>
> Our model generates feasible subgoals because it is trained on feasible subgoals: the training data consists of predicting an intermediate frame in a real trajectory that corresponds to the desired task. We do not do anything special to pick out “important” subgoals, so it is quite possible the model will produce “less important transitionary scenes” – that is a very valid point. However, we empirically observe that the model learns to generate useful subgoals from this training alone and the generated subgoals are sufficient to improve the performance of the downstream low-level controller, as evidenced by the good overall performance of our method in the real-world evaluations. We chose $k$, the horizon for the subgoals, by inspecting each dataset and choosing a value that will be about one-third of the average length of the trajectories in that dataset. For example, in the case of BridgeData, since most of the trajectories are around 20-40 steps, we set $k$ to 10. In the case of the CALVIN dataset, as most of the demos are 65 steps, we set $k$ to 20. Though simple, we found this choice to generally lead to good results. We do agree that a smarter way of adaptively choosing subgoals could lead to even better results. We have added this clarification in the updated paper.
> ___
>
> > **Table 1 uses some success rates from another paper (Ge et al., 2023). However, the approach proposed in this cited paper achieves higher success rates**
>
> We believe that the reviewer is referring to the non-zero-shot evaluation from Ge et al. 2023, which assumes access to interaction data from environment D (the target environment). All of the other methods we report, including ours, are performing zero-shot transfer, using the training environments (A, B, C) and without ever training on any trials from environment D. We believe it is reasonable to include only the zero-shot evaluations, and we’ve clarified this in the paper.
>
> That said, our updated CALVIN numbers in the revised draft (0.87, 0.69, 0.49, 0.38, 0.26) also substantially outperform this omitted result from Ge et al., 2023 (0.72, 0.47, 0.30, 0.13, 0.11). After the submission deadline, we realized that we used sub-optimal  values of classifier-free guidance parameters for the SuSIE evaluations on CALVIN. Simply tuning this value led to the currently reported result. The updated table can be seen at [this link](https://ibb.co/bJnJcPL).
>
> ___
>
> > **It was mentioned that Scene C should be a more difficult task than Scene B, however the results in 5.3 show that both MOO, RT-2-X and SuSIE perform significantly better on scene C than on B. Can you explain what the reason might be?**
>
>
> We apologize for the confusion. *Semantically*, Scene C is the most challenging since the robot needs to carefully ground the language command to identify the unseen objects and requires a high generalization capability. However, each object is in a shape that is relatively easy to grasp. In contrast, Scene B is challenging in the sense of manipulating the plastic bell pepper — this object requires particular precision to grasp due to being light, slippery, and almost as wide as the gripper. As such, because of these differences, it is unclear if absolute performances of different methods should be compared across these scenes. We have updated Section 5.1 in the new PDF to clarify this point.
> ___
>
> > **Would the robot in this pipeline be able to pick up such challenging objects, such as a heavy and slippery object, even after a few tries?**
>
> This is a good question. As discussed in the previous question, the bell pepper in Scene B is a challenging object to pick up – being light and slippery. We found that the subgoal generated by SuSIE improves the precision of low-level control for manipulating the challenging objects, as we discussed in Section 5.4. We also show qualitative videos on the [website](https://subgoal-image-editing.github.io/) under “Enhanced Precision -> Comparison Videos.”

---

> ### Author Response · Authors · 2023-11-19
> **Comment by Authors (Part 2)**
>
> ___
>
> > **InstructPix2Pix is not capable of performing viewpoint changes, how would this challenge be reflected in a real-life or research scenario where there are often multiple or moving cameras present?**
>
> While solving tasks involving changes in viewpoint is beyond our current focus, we believe this challenge can be addressed by training on more video data (e.g., Ego4D or Open X-Embodiment) that includes multiple or moving cameras. Another potential solution is to condition InstructPix2Pix on both historical and current observations. However, this is outside the scope of our work. We’ve added a discussion of this limitation of our evaluation setup to Section 6.
> ___
>
> > **The gripper may be obscured in the camera view due to other objects. How would the scene play out, would the robot be able to recover or get stuck?**
>
> In our robot setup, the gripper will never be completely obscured by obstacles. As discussed in the previous question, while this is beyond our focus, a potential solution in such partially observable tasks is to condition InstructPix2Pix on both historical and current observations, or add multiple cameras to the setup.
> ___
>
> > **What happens when it encounters, say, a similar current state but a different subgoal (that was not in the dataset) during test time?**
>
> We believe that our evaluation in Scene B already follows a similar setup to answer your question – this scene has a seen tabletop, a seen object (bell pepper), and a combination of a seen container (orange pot) and an unseen container (ceramic bowl). This means that while the initial state is similar to the state in the dataset, for the task “put the bell pepper in the ceramic bowl,” our method must produce a correct out-of-distribution subgoal. Our experiments demonstrate that our diffusion model can not only generate reasonable subgoals, but the low-level policy can also effectively handle these out-of-distribution subgoals.
> ___
>
> > **Does your robot dataset consist of only completely successful trajectories?**
>
> Yes, our datasets consist of successful trajectories. It will be an interesting direction to utilize negative trajectories and failure cases, which we leave to future work.
> ___
>
> > **There is a minor error in the table references. In Section 5.3 and A2.2.2, you mentioned "Table 5.3," which refers to "Table 1."**
>
> Thanks for the catch! We have fixed the reference in our updated PDF.
> ___
>
> > **Some parts could be reformulated/reworded better in Section 4**
>
> Thank you for pointing this out. We have now reformulated the notation in Section 4.1 in the updated PDF.
> ___
>
> > **The information regarding the datasets used can be provided more comprehensively and at an earlier stage in the paper.**
>
> We wanted to avoid excessively repeating information that is covered in a prior BridgeData V2 paper (Walke et. al., 2023), but have now added additional information about datasets in Appendix A.7. We are happy to move these details to the main paper for the final version, where there would be an extra page available.

---

> > ### Comment · Reviewer_rx11 · 2023-11-21
> > **Appreciated the explanations**
> >
> > Thank you for your explanations. The paper cannot cover all aspects of robotic manipulation in detail but I still believe there are some highly constraining assumptions made, e.g., gripper always visible, feasible subgoal prediction at all times, no viewpoint changes, independence of the diffusion model and the low-level policy, etc. Some of those points could have been investigated a little bit further.
> > Your clarifications helped me better understand some of the details, and I will consider them in my final evaluation.
> > On a final note, I hope you will share your code for full reproducibility.

---

> > > ### Author Response · Authors · 2023-11-21
> > >
> > > Hi, thanks for your response and positive feedback! We would like to clarify the remaining concerns you pointed out.
> > >
> > > **We hope these resolve your concerns, and we would be grateful if you are willing to upgrade your score!**
> > >
> > > ___
> > >
> > > > **gripper always visible, no viewpoint changes**
> > >
> > > We think these assumptions come from the robotic dataset we use, not from our method, as most of the robotic manipulation datasets are collected with fixed cameras and the gripper being visible (e.g. [MT-Opt](https://karolhausman.github.io/mt-opt/), [BridgeData](https://rail-berkeley.github.io/bridgedata/), [RoboTurk](https://roboturk.stanford.edu/),  [LanguageTable](https://interactive-language.github.io/), [Func Manipulation](https://sites.google.com/berkeley.edu/fanuc-manipulation), and more). It is worth noting that we have adopted common assumptions as seen in other recent works that were published at top ML and robotics conferences such as ICLR, ICML, CoRL, and RSS [1,2,3,4,5,6].
> > > ___
> > >
> > > >**feasible subgoal prediction at all times**
> > >
> > > Although there is no explicit detection for the feasibility of each subgoal, we find that the replanning mechanism at test-time execution (Algorithm 1) makes SuSIE surprisingly adept at recovering from failures – as our algorithm regenerates a new subgoal every K steps, even if an incorrect subgoal is accidentally generated, it can still successfully achieve the task as long as it generates the next subgoal correctly. You can find some qualitative examples of failure recovery at [this link](https://ibb.co/b61yQ4F). We have added this discussion to the paper.
> > > ___
> > >
> > > >**independence of the diffusion model and the low-level policy**
> > >
> > > There might be a misunderstanding here. We believe this assumption is not highly constraining, but rather it makes our method more flexible. Separating the training procedures gives us greater flexibility and reduces computational costs. Furthermore, these models are not entirely independent, as both are trained on the same robot dataset. Our experimental results demonstrate that our current recipe can lead to sufficiently good performance in downstream tasks.
> > > ___
> > >
> > > >**I hope you will share your code for full reproducibility**
> > >
> > > Yes, we are ready to release our code and pre-trained models to ensure full reproducibility! For reference, we have attached our anonymized code for training SuSIE’s diffusion model in the supplementary materials. We will also release the pre-trained checkpoint for our final public version!
> > > ___
> > >
> > > References
> > >
> > > [1] Ma et al., VIP: Towards Universal Visual Reward and Representation via Value-Implicit Pre-Training, ICLR 2023.
> > >
> > > [2] Ma et al., LIV: Language-Image Representations and Rewards for Robotic Control, ICML 2023.
> > >
> > > [3] Kumar et al., Pre-Training for Robots: Offline RL Enables Learning New Tasks from a Handful of Trials, RSS 2023.
> > >
> > > [4] Chen et al., GenAug: Retargeting behaviors to unseen situations via Generative Augmentation, RSS 2023.
> > >
> > > [5] Myers et al., Goal Representations for Instruction Following: A Semi-Supervised Language Interface to Control, CoRL 2023.
> > >
> > > [6] Walke et al., BridgeData V2: A Dataset for Robot Learning at Scale, CoRL 2023

---

> > > > ### Author Response · Authors · 2023-11-23
> > > >
> > > > Hi, thanks again for your positive feedback and the meaningful discussion. As the author-reviewer discussion period draws to the close in around 10 hours, we were hoping to check in with you if you have gotten a chance to go over our latest responses and if your remaining concerns are addressed. **If so, we would be grateful if you are willing to upgrade your score. We are happy to discuss further! Thanks!**

---

### Official Review · Reviewer_UFZY · 2023-10-31

**Soundness:** 2 fair
**Presentation:** 3 good
**Contribution:** 2 fair
**Rating:** 8
**Confidence:** 4

**Summary:**

This paper tackles the problem of zero-shot generalization to novel environments, objects, and scenarios in the context of robotics dexterous manipulation. The tasks are defined by a starting and end states, provided as visual observations. The goal is then to control a robotic arm to reach the goal state from the start state.

Authors propose to leverage large pre-trained language-guided image editing diffusion models to hallucinate the next subgoal or waypoint to reach to complete a given task. The diffusion model is thus used as a high-level planner whose goal is to output a close next sub-goal to find. A low-level policy is also trained to reach a sub-goal given its current state.

The diffusion model is fine-tuned on robotic data to learn to predict a waypoint given a language intersection, while the low-level policy is trained with behavior cloning on a dataset of trajectories.

The main claim of the paper is that the knowledge coming from Internet-scale pre-training of diffusion models allows a broader generalization to new scenarios and that such models are able to provide physically realistic new sub-goals. As all high-level planning is done by the generative model, the policy only has to perform low-level short-term control to reach a close waypoint, leading to more precise manipulation.

Experiments are conducted both in simulation and on a real-world robotic platform, and the introduced method is compared against diverse relevant baselines.

**Strengths:**

* **S1**: The idea is straightforward and properly motivated in the paper. Indeed, defining two modules, i.e. high-level planning and low-level control, appears to be an interesting idea to simplify the task to learn. Moreover, framing high-level planning as an image editing task and focusing on only predicting the short-term future (instead of full video prediction) is an appealing approach.

* **S2**: The paper is well-written and easy to follow.

* **S3**: Experiments are conducted both in simulation and on a real-world platform, and the proposed method is compared against several challenging baselines.

**Weaknesses:**

* **W1**: **[Major]** Quality of the predicted sub-goals: Authors present a qualitative study of predicted sub-goals in Figure 3. They claim that the predicted sub-goal images are physically plausible and of higher quality when leveraging the *InstructPix2Pix* pre-training. I find it hard to assess the quality of the generated images only from looking at these examples. This leads to 2 concerns:
    * **W1.1**: Is the pre-training improving the downstream policy performance? The paper only presents a qualitative comparison of predicted sub-goals. Authors should report quantitative results, controlling the low-level policy to reach subgoals generated by the 2 models (with and without InstructPix2Pix pre-training) to show that the Internet-scale pre-training brings a boost in performance.
    * **W1.2**: Are the predicted subgoals really physically plausible, and is it a pre-requisite for the low-level policy to succeed in controlling the robotic arm? By looking at Figure 3, it is not easy to verify that sub-goals are physically plausible. Further experiments should be conducted to verify this quantitatively. Moreover, authors should conduct experiments to show that having physically plausible sub-goals improves the performance of the low-level policy. A possible scenario might be that sub-goals are not always physically plausible, but through its training, the low-level policy has learned to come as close as possible to sub-goals while avoiding ending up in any undesirable state.

* **W2**: **[Major]** How are findings from Figure 4 generalizable? It shows qualitative rollouts from the proposed method and RT-2-X which seem to indicate that RT-2-X manipulates objects with less precision. However, quantitative results in Table 2 show that RT-2-X outperforms the introduced method in certain tasks, showing it might be more precise in some cases.

* **W3**: **[Major]** RT-2-X outperforms the method on certain tasks. However, as mentioned by the authors, RT-2-X was trained on a large robotics dataset and the proposed method still showcases promising performance on most tasks. I thus believe this smaller performance compared with RT-2-X on scene C does not outweigh the interesting contributions of this work.

* **W4**: **[Minor]** To the best of my knowledge, the policy action space is not described in the main paper. A description should be added.

* **W5**: **[Minor]** Even if authors properly reference the paper of interest and refer to the appendix later, the following sentence is quite vague: “To obtain a robust goal-reaching policy in Phase (ii), we follow the implementation details prescribed by Walke et al. (2023).” Additional details should be added directly in the main paper following this sentence.

**Questions:**

All questions and suggestions are already mentioned in the “Weaknesses” section as a list of numbered points.

---

> ### Author Response · Authors · 2023-11-19
>
> Thank you for your feedback and a positive assessment of our work. We added new experimental results to compare SuSIE to the model without pretraining, and also included clarifications about the physical plausibility and low-level precision.
>
> **Please let us know if your questions are resolved, and if so, we would be grateful if you are willing to upgrade your score. We are happy to discuss further if any questions are remaining.**
> ___
>
> > **W1.1: Is the pre-training improving the downstream policy performance?**
>
> This is a good question! We have conducted an additional real robot experiment evaluating the from-scratch model in Scene C. As shown in the table, the from-scratch model is completely unable to perform the task, which demonstrates that the InstructPix2Pix initialization is essential for robust generalization. The videos of the from-scratch model can be found at [this link](https://ibb.co/XXrxFbt), including the generated subgoals in the top row. More additional qualitative examples can be found at [this link](https://ibb.co/4fR6NQk) showing that in unseen environments, the model without InstructPix2Pix initialization (the column “from scratch”) produces noticeably worse images.
>
> | Task | From Scratch | Ours |
> | -------- | -------- | -------- |
> | Toothpaste in bowl | 0.0     | 0.5     |
> | Crayon in bowl      | 0.0     | 0.6     |
> | Spoon in bowl      | 0.0     | 0.4     |
> | Bowl to top      | 0.0     | 0.3     |
> ___
>
> > **W1.2: Are the predicted subgoals really physically plausible? Authors should conduct experiments to show that having physically plausible sub-goals improves the performance of the low-level policy.**
>
> We apologize for the confusion. While we do not explicitly evaluate the plausibility of a given subgoal, we found that the predicted intermediate subgoals tend to provide more fine-grained feedback on gripper orientations and pose at the intermediate steps of a trajectory, which allows the policy to be more precise. The previous comparison with the from-scratch model empirically supports this claim –  while the from-scratch model generates implausible subgoals and is useless for performing the task, the subgoals generated by SuSIE are sufficiently plausible for effective low-level control.
>
> In addition, to further investigate if having plausible sub-goals improves the performance, we conducted a comprehensive comparison with Oracle GCBC (GCBC with the ground-truth final goal). SuSIE constantly achieves higher success rates even compared to executing privileged ground-truth final goals as shown in [this table](https://ibb.co/x8gDN7M). We also show qualitative videos on  [our website](https://subgoal-image-editing.github.io/) under “Enhanced Precision -> Comparison Videos.”
>
> We are happy to further revise the wording in the paper if that'd be helpful to improve clarity.
> ___
>
> > **W2: [Major] How are findings from Figure 4 generalizable? W3: [Major] RT-2-X outperforms the method on certain tasks.**
>
> After the submission deadline, we made some improvements to the subgoal diffusion model as we discuss in Appendix A.4. We re-evaluated in all scenes and found that our method now outperforms RT-2-X, even in Scene C. You can find the updated results at [this link](https://ibb.co/hKDHFFB), as well as in Table 2 in the updated PDF.
>
> In addition, to further support our results presented in Figure 4, we conducted a comprehensive comparison with Oracle GCBC as we showed in the previous question. These results demonstrate that the findings from Figure 4 are generalizable.
> ___
>
> > **W4: [Minor] The policy action space is not described in the main paper.**
>
> The action space is seven-dimensional, consisting of a 6D relative Cartesian motion of the end-effector, and a discrete dimension to control the gripper action. We have added the description in Section 4.3 in the updated PDF.
> ___
>
> > **W5: [Minor] Additional details should be added directly in the main paper following this sentence.**
>
> Due to the page limit, we describe the details of the goal-reaching policy in Appendix A.1.1. We will include the description in the main paper for our final version (which will have an extra page for).

---

> ### Author Response · Authors · 2023-11-21
> **A friendly reminder**
>
> Dear Reviewer UFZY,
>
> Thank you for your feedback! We were wondering if you have gotten a chance to go over our responses. We would appreciate it if you could have a look and let us know if your questions are resolved. We are happy to discuss further.
>
> Thanks!

---

> > ### Comment · Reviewer_UFZY · 2023-11-21
> >
> > I thank the authors for their answers to my different questions, along with additional experiments and results.
> >
> > Provided experiments address my concerns and clarifications make the contribution clearer. I am thus increasing my overall score.

---

> > > ### Author Response · Authors · 2023-11-23
> > >
> > > Thank you for increasing your score! We greatly appreciate the comments and feedback from the reviewer, which have largely helped us improve our paper. Thanks a lot for your effort and time!

---

### Official Review · Reviewer_rGNF · 2023-11-06

**Soundness:** 2 fair
**Presentation:** 4 excellent
**Contribution:** 2 fair
**Rating:** 5
**Confidence:** 4

**Summary:**

SuSIE enables robots to recognize and reason about new objects and scenarios using a pre-trained image editing diffusion model, InstructPix2Pix. It proposes intermediate subgoals for the robot to reach, which are fine-tuned on robot data. A low-level policy is trained to reach these subgoals, resulting in robust generalization capabilities. The approach successfully solves real robot control tasks with novel objects, distractors, and environments in both real world and simulation settings.

**Strengths:**

SuSIE uses a pre-trained image editing diffusion model as a high-level planner, along with a low-level goal-conditioned policy trained to achieve these subgoals.
The approach successfully solves real robot control tasks that involve new objects, distractions, and environments.
The method is compared to existing approaches in terms of incorporating semantic information from pre-trained models.
The paper is well-written and easy-to-read.

**Weaknesses:**

The diffusion model and low-level policy are trained separately, meaning the diffusion model is unaware of the low-level policy's capabilities.
There is no discussion on how to handle the situation if the generated subgoal is incorrect, nor are there any mechanisms for error detection and recover.
Most of the tasks presented in the paper appear to be easily solvable using pre-trained perception modules and traditional planning methods.
The core of the method InstructPix2Pix can only infer from visual information and cannot infer any physical properties of the scene that certain tasks require.
Real-world evaluation is conducted in just three scenes, which may not fully capture the complexity and variability of actual manipulation tasks.

**Questions:**

1. In Algorithm 1
I was wondering how to determine the key frames that cannot be described in a single instruction. You mentioned sampling a new subgoal every K steps. How do you choose the goal sampling interval K to ensure the subgoals are plausible enough?
2. In 5.2 you mention "an understanding of dynamics to predict how to move and rotate the gripper to grasp it."
How can you determine the plausibility of the prediction in terms of dynamics, given that the generating subgoals and learning low-level control policies are separate?
3. It seems that an extremely accurate pose of the end effectors is needed. I am somewhat concerned that the precision of the genarated subgoal frame may not be sufficient to achieve it.

---

> ### Author Response · Authors · 2023-11-19
> **Comment by Authors (Part 1)**
>
> Thank you for your detailed feedback! To address the issues raised in your review, we added new experiments to show the physical plausibility and precisions of subgoals and also added clarification regarding error detection and recovery, difficulty of tasks, and goal sampling interval K.
>
> **Please let us know if your questions are resolved, and if so, we would be grateful if you are willing to upgrade your score. We are happy to discuss further if any questions are remaining.**
> ___
>
> > **There is no discussion on how to handle the situation if the generated subgoal is incorrect, nor are there any mechanisms for error detection and recover**
>
> Although there is no explicit error detection, we find that the replanning mechanism at test-time execution (Algorithm 1) makes SuSIE surprisingly adept at recovering from failures – as our algorithm regenerates a new subgoal every K steps, even if an incorrect subgoal is accidentally generated, it can still successfully achieve the task as long as it generates the next subgoal correctly. You can find some qualitative examples of failure recovery at [this link](https://ibb.co/b61yQ4F), as well as more on [our website](https://subgoal-image-editing.github.io/) under “Enhanced Precision -> Comparison Videos.” We have added this discussion to the paper.
> ___
>
> > **Most of the tasks presented in the paper appear to be easily solvable. Real-world evaluation is conducted in just three scenes.**
>
> We evaluated a total of **9 tasks across three different scenes**, and we believe that the difficulty and number of tasks in our evaluation are sufficient to answer the 4 research questions we pose at the beginning of Section 5. It is worth noting that we have adopted a common evaluation scope, as seen in other recent works on data-driven robot learning, that were also published at top ML and robotics conferences such as ICLR, ICML, CoRL, and RSS [1,2,3,4,5,6,7].
>
> [1] studies only simulation tasks and did not include any real-world evaluations. [2] only evaluated 1 single tabletop scene for 4 tasks on the real robot. [3] also studies only 1 single scene for 9 tasks, and all tasks are picking and placing. [4] considers 3 scenes, which consist of 2 seen scenes in the training dataset. [5] studies 3 scenes, but all scenes look similar and have less variability than ours. [6] and [7] study a similar number of scenes as ours, consisting largely of picking and placing on a tabletop.
>
> [1] Yu et al., ​​Using Both Demonstrations and Language Instructions to Efficiently Learn Robotic Tasks, ICLR 2023.
>
> [2] Ma et al., VIP: Towards Universal Visual Reward and Representation via Value-Implicit Pre-Training, ICLR 2023.
>
> [3] Ma et al., LIV: Language-Image Representations and Rewards for Robotic Control, ICML 2023.
>
> [4] Kumar et al., Pre-Training for Robots: Offline RL Enables Learning New Tasks from a Handful of Trials, RSS 2023.
>
> [5] Myers et al., Goal Representations for Instruction Following: A Semi-Supervised Language Interface to Control, CoRL 2023.
>
> [6] Walke et al., BridgeData V2: A Dataset for Robot Learning at Scale, CoRL 2023.
>
> [7] Stone et al., Open-World Object Manipulation using Pre-Trained Vision-Language Models, CoRL 2023.

---

> ### Author Response · Authors · 2023-11-19
> **Comment by Authors (Part 2)**
>
> ___
> > **How do you choose the goal sampling interval K to ensure the subgoals are plausible enough?**
>
> While a full ablation of this hyperparameter would be ideal, retraining the diffusion model and re-evaluating the policy many times is prohibitively expensive. As such, we chose $k$ by inspecting each dataset and choosing a value that will be about one-third of the average length of the trajectories in that dataset. For example, in the case of BridgeData, since most of the trajectories are around 20-40 steps, we set $k$ to 10. In the case of the CALVIN dataset, as most of the demos are 65 steps, we set $k$ to 20. Though simple, we found this choice to generally lead to good results. We do agree that a smarter way of adaptively choosing subgoals could lead to even better results.
> ___
>
> > **In 5.2 you mention "an understanding of dynamics to predict how to move and rotate the gripper to grasp it." How can you determine the plausibility of the prediction in terms of dynamics, given that the generating subgoals and learning low-level control policies are separate?**
>
> We apologize for the confusion. We have revised this sentence in the updated PDF. While we do not explicitly evaluate the plausibility in terms of the dynamics of a given subgoal, we found that the predicted intermediate subgoals tend to provide more fine-grained feedback on gripper orientations and pose at the intermediate steps of a trajectory, which allows the policy to be more precise, and this is what we meant by this sentence.
>
> In addition, we found that pre-training is crucial for our model to generate fine-grained subgoals for the low-level policy to handle. We conducted an additional experiment to compare SuSIE to the model without InstructPix2Pix initialization (trained from scratch on robot data). As shown in the table, the from-scratch model generates implausible subgoals and is completely unable to perform the task. This demonstrates the fact that the subgoals generated by SuSIE are sufficiently plausible for the low-level control. The qualitative videos of the from-scratch model can be found at [this link](https://ibb.co/XXrxFbt), including the generated subgoals in the top row.
>
> | Task | From Scratch | Ours |
> | -------- | -------- | -------- |
> | Toothpaste in bowl | 0.0     | 0.5     |
> | Crayon in bowl      | 0.0     | 0.6     |
> | Spoon in bowl      | 0.0     | 0.4     |
> | Bowl to top      | 0.0     | 0.3     |
>
> In short, while explicit detection of subgoal plausibility will likely improve the performance further, our experimental results show that our current system, although simple, is already capable of effectively performing the tasks. We are happy to further revise this wording if that'd be helpful to improve clarity.
>
> ___
>
>
> > **It seems that an extremely accurate pose of the end effectors is needed. I am somewhat concerned that the precision of the generated subgoal frame may not be sufficient to achieve it.**
>
> As mentioned above, whether or not the subgoals are precise enough is an empirical question that we answer with our experiments. Our comparison with Oracle GCBC (Section 5.4) demonstrates that SuSIE consistently achieves higher success rates even compared to executing privileged ground-truth final goals. In addition, we further compared SuSIE with Oracle GCBC in all scenes and the results can be found at [this link](https://ibb.co/x8gDN7M). Qualitative examples are provided in Figure 4 as well as on [our website](https://subgoal-image-editing.github.io/) under “Enhanced Precision -> Comparison Videos.” These results demonstrate that the subgoals generated by SuSIE are sufficiently accurate to improve the precision of low-level control. Generally, we do not believe that it is necessary to have particularly accurate poses for the end effector, nor did we observe this to be a significant factor – perhaps this stems from some misunderstanding of the method? Could you clarify why you believe that the end effector poses need to be highly precise? If this is due to confusing writing on our part we would like to correct it!

---

> ### Author Response · Authors · 2023-11-21
> **A friendly reminder**
>
> Dear Reviewer rGNF,
>
> Thank you for your feedback! We were wondering if you have gotten a chance to go over our responses. We would appreciate it if you could have a look and let us know if your questions are resolved. We are happy to discuss further.
>
> Thanks!

---

> > ### Author Response · Authors · 2023-11-23
> > **Discussion period closing soon**
> >
> > Dear Reviewer rGNF,
> >
> > As the author-reviewer discussion period draws to the close in around 10 hours, we were hoping to check in with you if you have gotten a chance to go over our responses and if your concerns are addressed. We have performed several additional experiments and would be grateful if you engage in a discussion with us. We are happy to address any remaining concerns. Thank you so much!

---

> > > ### Comment · Reviewer_rGNF · 2023-11-23
> > >
> > > Thank you for your response. I respect your perspective and appreciate the effort put into it. However, I still perceive this paper as borderline. Consequently, I'll maintain my initial ranking and look forward to the other reviewers' assessments to finalize the decision.

---

> > > > ### Author Response · Authors · 2023-11-23
> > > >
> > > > Thank you for your response. We really appreciate your effort and are grateful for your willingness to consider assessments of other reviewers in finalizing your decision. Through the discussion period, other reviewers have generally found their concerns to be mostly addressed.
> > > >
> > > > We believe our responses should have addressed the concerns you pointed out, but we are very happy to address any remaining concerns in the remainder of the discussion period (as much as possible within the time) and, definitely, in the final version of the paper. It would be awesome if you could specify your remaining concerns. Thanks again for providing feedback, for engaging in a discussion with us, and for your willingness to reconsider your score.

---

### Comment · Area_Chair_fXfc · 2023-11-20
**Author-Reviewer Discussion Period Ending November 22**

Hi,

Thanks for your help with the review process!

There are only two days remaining for the author-reviewer discussion (November 22nd). Please read through the authors' response to your review and comment on the extent to which it addresses your questions/concerns.

Best,\
AC

---

### Meta-Review · Area_Chair_fXfc · 2023-12-12

**Metareview:**

The paper proposes SuSIE, a method that utilizes a language-guided image-editing diffusion model as a high-level planner that generates subgoals for a low-level robot control policy. The diffusion model is an InstructPix2Pix model that is fine-tuned on robot data to generate (hallucinate) possible future observations (subgoals) according to language and the current image. The low-level policy is then trained to reach these subgoals. The method is evaluated in simulation as well as on a few real-world robot manipulation tasks, and is shown to generalize to novel objects, environments, and distractors.

The paper was reviewed by four referees who largely agree on the strengths and limitations of the paper. Among the strengths, at least three reviewers emphasize that the paper is well organized, well written, and easy to follow. They find the method to be straightforward and intuitive, and to be well motivated. They also comment on the soundness of the experimental setup. A few reviewers raised concerns about possible inconsistencies in the comparison with the RT-2-X baseline, the sufficiency of the real-world experiments, and the nature of the results on CALVIN. As acknowledged by the reviewers, the authors made a concerted effort to address these concerns, in part by presenting new experimental results and by resolving issues with previous results. While the AC appreciates reference to the experimental evaluation in previous ICLR/ICML papers as justification for the sufficiency of the real-world results, it is not unreasonable to ask that the paper provide more extensive real-world results, particularly given the stated goal of realizing generalist robots that are able to operate in truly unstructured environments.

**Justification For Why Not Higher Score:**

While the objectives are of interest to many in the robot learning community, the significance of the algorithmic contributions are not sufficient to warrant a higher rating.

**Justification For Why Not Lower Score:**

The approach is interesting and intuitive.

---

### Decision · Program_Chairs · 2024-01-16

Accept (poster)